# Coordination of siderophore gene expression among clonal cells of the bacterium *Pseudomonas aeruginosa*

Subham Mridha [1✉] & Rolf Kümmerli [1✉]

There has been great progress in understanding how bacterial groups coordinate social actions, such as biofilm formation and public-goods secretion. Less clear is whether the seemingly coordinated group-level responses actually mirror what individual cells do. Here, we use a microscopy approach to simultaneously quantify the investment of individual cells of the bacterium *Pseudomonas aeruginosa* into two public goods, the siderophores pyochelin and pyoverdine. Using gene expression as a proxy for investment, we initially observe no coordination but high heterogeneity and bimodality in siderophore investment across cells. With increasing cell density, gene expression becomes more homogenized across cells, accompanied by a moderate shift from pyochelin to pyoverdine expression. We find positive associations in the expression of pyochelin and pyoverdine genes across cells, with cell-to-cell variation correlating with cellular metabolic states. Our work suggests that siderophore-mediated signalling aligns behaviour of individuals over time and spurs a coordinated three-phase siderophore investment cycle.

[1] Department of Quantitative Biomedicine, University of Zürich, Winterthurerstrasse 190, 8057 Zürich, Switzerland. ✉email: subhammridha@yahoo.com; rolf.kuemmerli@uzh.ch

Bacteria perform many actions that are vital for survival and growth outside their cells[1]. For example, bacteria secrete enzymes to digest extra-cellular polymers and proteins, siderophores to scavenge iron, and biosurfactants to enable swarming on wet surfaces[2–5]. A consequence of molecule secretion is that the benefit of the performed actions can accrue to other cells, because digested nutrients, iron-loaded siderophores, and biosurfactants become accessible to other individuals within a group[6–9]. Because of their group-level effects, molecule production often occurs in a coordinated fashion, relying on molecular mechanisms such as quorum sensing and other signalling circuits, integrating information across individuals within a group[10–12]. There has been tremendous interest in understanding the molecular basis of these regulatory circuits, and the ecological and evolutionary consequences of coordinated molecule production[5,13–15]. Yet, many of the insights gained on group-coordinated behaviour originate from batch-culture experiments, where behavioural responses were averaged across millions of cells. In contrast, we still know little on how individual cells behave within a group, and whether decisions are taken by individuals indeed match the patterns we interpret as coordinated group response at the population level[16–18].

Here, we tackle these open issues by studying single-cell social behaviour in the opportunistic human pathogen *Pseudomonas aeruginosa* PAO1. This species produces a suit of extra-cellular compounds that can benefit other group members[10]. In our study, we focus on the investment strategies of individual cells into the two iron-chelating siderophores pyoverdine and pyochelin. Siderophores are produced and secreted in response to iron limitation in order to scavenge this essential nutrient from the environment[19,20]. Siderophore-iron complexes are subsequently taken up by cells via cognate receptors[21]. Due to their extra-cellular mode of action and their high diffusivity, siderophores can be considered 'public goods', benefiting other cells in a clonal group[5].

*P. aeruginosa* produces pyochelin and pyoverdine via non-ribosomal peptide synthesis[22]. The regulation of these two siderophores involves three main levels. The first level is mediated by Fur (ferric uptake regulator). Fur blocks siderophore synthesis, but loses its inhibitory effect when intra-cellular iron stocks become depleted[20,23–25]. Fur de-repression initiates basal siderophore gene expression, with evidence existing that de-repression happens earlier for pyochelin than for pyoverdine[26]. The second level involves signalling cascades, where incoming siderophore-iron complexes trigger positive feedback loops that increase siderophore production[27–31]. For pyoverdine, the pathway involves two sigma factors PvdS and FpvI. While PvdS induces the expression of pyoverdine synthesis and export genes, FpvI induces the expression of the outer membrane receptor FpvA, required for ferri-pyoverdine uptake. The sigma factors PvdS and FpvI are bound and inhibited by the anti-sigma factor FpvR which spans the cytoplasmic membrane. The import of ferri-pyoverdine via FpvA results in the proteolytic degradation of FpvR, which releases PvdS and FpvI and thereby fully induces the pyoverdine pathway[27,28]. For pyochelin, the positive feedback directly involves the iron-loaded siderophore, which is imported into the cell via the outer membrane receptor FptA and the permease FptX. In the cytosol, ferri-pyochelin activates the transcription factor PchR (belonging to the AraC-type family), which in turn guides the expression of pyochelin synthesis, export, and import genes[29–31]. The third level is based on a hierarchical regulatory linkage between the two siderophores, whereby pyoverdine suppresses pyochelin production within cells[26,32]. While the exact mechanism of repression is unknown, the most likely explanation is that pyoverdine with its high iron affinity ($K_a = 10^{32}\,M^{-1}$) reduces the complexation of iron with the low affinity pyochelin ($K_a = 10^{18}\,M^{-2}$)[33,34], thereby inhibiting pyochelin-signalling and synthesis. Such hierarchical regulation of siderophores has been described in a number of species[35,36]. In addition to these three main levels, further regulatory elements that also influence siderophore production have been described[37–41]. We will discuss their potential role on influencing inter-individual variation later on in the manuscript.

Important to note is that all the three main levels interact and there is likely heterogeneity in the relative strength of each level across cells in a clonal population. For one thing, cells might vary in their intra-cellular iron stocks, which affects the strength of Fur repression/de-repression[23] and in turn could determine whether a cell primarily invests in pyochelin, pyoverdine, or none of the siderophores[26]. Furthermore, the extracellular iron availability and local cell density each individual experiences likely varies between cells, which will influence the uptake rate of siderophore-iron complexes and thus the strength of the positive feedbacks. Moreover, it is reasonable to assume that the regulatory mechanisms underlying the three input levels are intrinsically noisy themselves[17]. Given all these sources of variation, a key question is how groups of cells can coordinate and fine-tune their siderophore production strategy. One extreme answer could be that there is no coordination and that the observed population level responses[42–44] might merely be the sum of its heterogenous individual members. At the other extreme, it could be that the regulatory circuits operate in a way that fosters specialization[45], where fractions of cells in a population invest either in pyochelin or pyoverdine.

Here, we examine these possibilities by using single-cell microscopy to simultaneously track the investment of individual cells into pyochelin and pyoverdine, across a range of media differing in iron availability, and across time following a growth cycle from low to high cell density. We used the expression of pyochelin and pyoverdine synthesis genes as proxies for siderophore investment levels and verified this assertion by comparing natural pyoverdine fluorescence to gene expression data. We grew bacteria in batch cultures for 24 h in seven different media and extracted small samples from these cultures at 3 h intervals to quantify the expression of pyochelin and pyoverdine genes in cells using fluorescent gene-reporter fusions. Crucially, we constructed double-fluorescent gene reporters (Figs. S1 and S2), which allowed us to simultaneously obtain single-cell investment proxies for both siderophores, and for the two siderophores and a housekeeping gene. This enabled us to establish gene expression correlation patterns across cells within populations.

## Results

**Population-level growth and siderophore gene expression**. We first quantified the growth and siderophore gene expression of *P. aeruginosa* at the population level, across a range of casamino acids (CAA) media compositions, differing in their level of iron limitation. As expected, we found that iron supplementation increased population growth, while the addition of iron chelators reduced growth relative to the plain CAA medium (Fig. 1a, b and Table S1). However, the two iron chelators used (bipyridyl and human apo-transferrin) varied in their effect on growth. The addition of bipyridyl predominantly extended the lag-phase of cultures (Fig. S3, Table S1, $F_{1,238} = 144.9$, $p < 0.0001$), but had only a mild negative effect on the growth integral ($F_{1,238} = 18.08$, $p < 0.0001$) and no significant effect on the maximal growth rate (Fig. S3, Table S1, $F_{1,238} = 1.65$, $p = 0.2002$). Conversely, the addition of transferrin significantly affected all three growth parameters (Fig. S3, Table S1, lag-phase: $F_{1,238} = 285.7$, $p < 0.0001$; growth integral: $F_{1,238} = 1292$, $p < 0.0001$; maximal growth rate: $F_{1,238} = 22.67$, $p < 0.0001$).

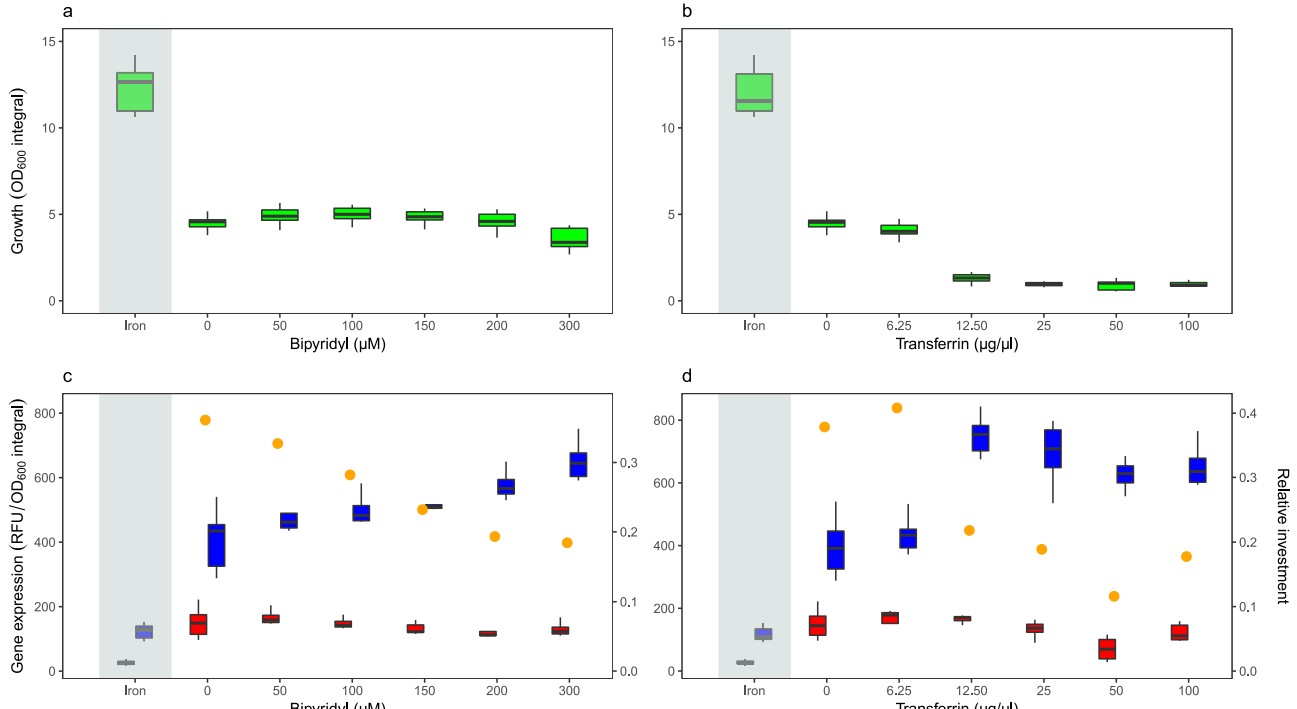

**Fig. 1 Higher levels of iron limitation reduce population growth and induce a relative increase of pyoverdine versus pyochelin gene expression in *P. aeruginosa*. a, b** Growth measured as integral (area under the OD600 curve) over 24 h batch culture experiments in iron-replete CAA medium (100 μM $FeCl_3$, grey shaded area) and CAA media with increasing concentrations of the iron chelators bipyridyl (**a**) and human apo-transferrin (**b**). Higher iron chelator concentrations in the medium significantly reduced bacterial growth (see Table S1 for statistics). **c, d** Siderophore reporter gene expression (red: pyochelin PAO1*pchEF:mcherry*; blue: pyoverdine PAO1*pvdA:mcherry*) measured as normalized integral (area under the relative fluorescence unit (RFU) curve divided by the OD600 integral) over 24 h batch culture experiments. Gene expression was measured in iron-replete CAA medium (100 μM $FeCl_3$, grey shaded area) and CAA media with increasing concentrations of the iron chelators bipyridyl (**c**) and human apo-transferrin (**d**). Relative gene expression was measured as the ratio of normalized pyochelin to pyoverdine expression (yellow dots). Boxplots represent the median with 25th and 75th percentiles, and whiskers show the 1.5 interquartile range.

With regard to gene expression, we found low fluorescence signals for pyochelin (PAO1*pchEF:mcherry*) and pyoverdine (PAO1*pvdA:mcherry*) in iron-supplemented medium (Fig. 1c, d), confirming that the investment in the two siderophores is reduced to baseline levels to save unnecessary costs[42,43,46]. Conversely, pyochelin and pyoverdine genes were expressed in plain CAA medium and CAA supplemented with iron chelators. While pyoverdine gene expression significantly increased with higher chelator concentrations (for bipyridyl: $F_{1,79} = 108.3$, $p < 0.0001$; for transferrin: $F_{1,78} = 64.52$, $p < 0.0001$; Fig. S4b, d), pyochelin gene expression significantly decreased (for bipyridyl: $F_{1,77} = 8.77$, $p = 0.0041$; for transferrin: $F_{1,78} = 17.96$, $p < 0.0001$; Fig. S4a, c). Consequently, the ratio of pyochelin-to-pyoverdine gene expression significantly changed in favour of pyoverdine with higher levels of iron limitation (for bipyridyl: $F_{1,4} = 162.4$, $p = 0.0002$; for transferrin: $F_{1,4} = 12.05$, $p = 0.0255$). Temporal dynamics of population-level gene expression confirmed these patterns (Fig. 2). Pyoverdine gene expression increased monotonously over time, while pyochelin gene expression plateaued or even declined over time with high chelator concentrations. Finally, we confirmed that gene expression fluorescence is an adequate proxy for actual siderophore production (Fig. S5) by comparing the natural green fluorescence of pyoverdine to the fluorescence of the gene expression reporter[47]. We found strong positive correlations between *pvdA::mcherry* gene expression and natural pyoverdine fluorescence in both media supplemented with bipyridyl (Fig. S5a) and apo-transferrin (Fig. S5b).

**Single-cell level siderophore gene expression across time and iron limitations**. We then used the double reporter PAO1*pvdA::mcherry–pchEF::egfp* to quantify pyochelin and pyoverdine gene expression of individual cells over a 24 h growth cycle in batch cultures, across seven different variants of the CAA medium differing in their level of iron availability. In the following sub-sections, we will dissect the complex expression patterns by first focusing on the iron-supplemented treatment, and then turning to iron-limited media supplemented with either bipyridyl or apo-transferrin. The single-cell data is shown in two different ways: as dot plots (Fig. 3, where each dot represents an individual cell) and as density plots (Fig. S6). Note that we repeated all experiments with the single gene reporters PAO1*pvdA:mcherry* and PAO1*pchEF:mcherry* as controls, which run in parallel with our double reporter. The results from the single (Fig. S7) and double reporters (Fig. 3) are congruent, demonstrating that our findings are neither influenced by the construct type nor the fluorophore (mCherry vs. GFP). Moreover, we correlated the *pvdA::mcherry* gene expression with the natural pyoverdine fluorescence at the single-cell level (Fig. S5c, d) to further confirm that gene expression is a valid proxy for siderophore investment. Indeed, we found strong positive correlations between the gene expression read-out and the actual pyoverdine content of cells at early time points (up to 6th hour). Later on, the correlations became weaker, which is expected because pyoverdine is secreted and shared between cells[48,49], such that the natural fluorescence of a cell reflects the sum of the self-produced pyoverdine and the uptake from other cells.

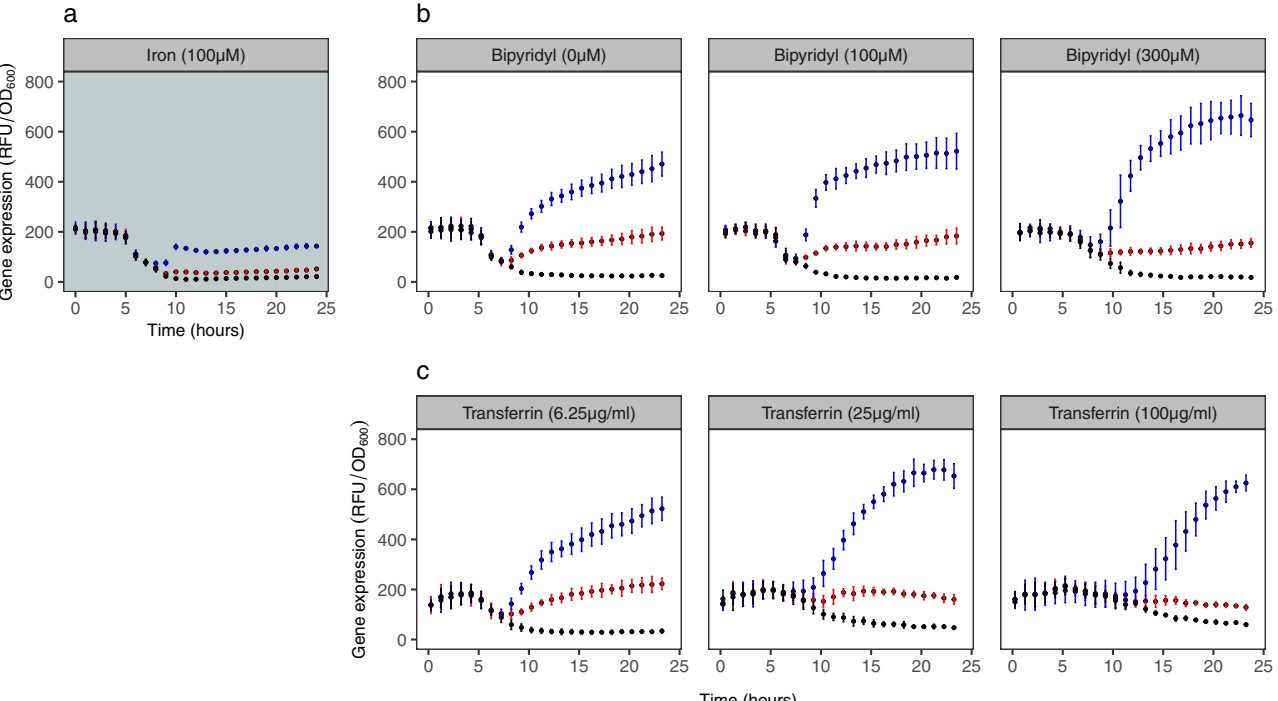

**Fig. 2 Temporal dynamics of siderophore reporter gene expression at the population level.** The panels show reporter gene expression for pyochelin (red: PAO1*pchEF:mcherry*), pyoverdine (blue: PAO1*pvdA:mcherry*), and the wildtype control strain (black: PAO1 without reporter fusion) across a range of CAA media differing in their level of iron limitations. Values represent normalized gene expression (relative fluorescence units (RFU) divided by OD600) in: **a** iron-replete CAA medium (100 μM FeCl₃); **b** CAA media with increasing concentrations of the iron chelator bipyridyl; **c** CAA media with increasing concentrations of the iron chelator apo-transferrin. Values and error bars represent the mean and standard deviation across 8 replicates (2 experiments with 4 replicates each), respectively. The inflation of normalized gene expression in the early hours of the growth can be attributed to divisions by very low OD600 values (close to zero) in this growth phase. Gene expression patterns started to segregate from the control PAO1 strain upon entering the exponential growth phase.

*Siderophore gene expression patterns in iron-rich medium.* Consistent with previous batch-cultures studies[19,23], we found that the average *pchEF* and *pvdA* gene expression did not differ from background levels in iron-rich CAA medium, regardless of the time points analyzed (Fig. 3a, b). However, gene expression varied widely between cells (Fig. 3a, b). For pyochelin, we observed bimodal gene expression up to 6 h, where a fraction of cells (16%) showed relatively high *pchEF* expression activity (log(fluo) ≥ 1), whereas the remaining cells were primarily in the off stage (Fig. S6a). At later time points, the bimodality disappeared. When comparing standard deviations as a proxy for gene expression heterogeneity (Fig. 4a), we observed that *pchEF* expression heterogeneity peaked early on during the experiment (3–6 h) and then declined afterwards. For pyoverdine, there was no bimodality in gene expression (Fig. S6b), but relatively high and time-consistent heterogeneity in *pvdA* gene expression across cells (Fig. 4a).

*Siderophore gene expression patterns in media supplemented with bipyridyl.* As for the iron-rich environment, we observed that gene expression varied widely between cells (Fig. 3c, d). For pyochelin, we consistently found bi-modal *pchEF* expression during the first six hours of the experiment, regardless of how much bipyridyl was added (Fig. S6c). Under all conditions, there was a high and a low proportion of cells investing marginally and highly in pyochelin gene expression, respectively. At later time points, gene expression levels across cells converged to intermediate levels (Fig. 3c). Across all three bipyridyl concentrations, we found that *pchEF* gene expression moderately but significantly declined from the 9th hour onwards (Fig. 3c). For pyoverdine, the

pattern was different in the sense that we observed a gradual induction of gene expression from an off-low to an all-on stage (Figs. 3d, S6d). Overall, we found that the heterogeneity in *pchEF* and *pvdA* expression significantly decreased over time ($F_{1,43} = 12.90$, $p = 0.0008$), with the decrease being similar for both genes ($F_{1,43} = 0.13$, $p = 0.7207$) and across media ($F_{1,43} = 0.61$, $p = 0.5500$; Fig. 4b).

*Siderophore gene expression patterns in media supplemented with apo-transferrin.* The single-cell gene expression patterns in CAA media supplemented with apo-transferrin mirrored at large the patterns observed in media containing bipyridyl: (i) heterogeneity in gene expression across cells was high, especially during the first six hours of the experiment (Figs. 3e, f, S5e, f); (ii) heterogeneity in *pchEF* and *pvdA* expression significantly decreased over time ($F_{1,43} = 26.93$, $p < 0.0001$), similarly for both genes ($F_{1,43} = 0.19$, $p = 0.6625$) and across media ($F_{1,48} = 2.59$, $p = 0.0865$; Fig. 4c); and (iii) *pchEF* gene expression declined moderately with time (Fig. 3e) at all apo-transferrin concentrations. However, there were also a few notable differences. First, bimodal gene expression during the first six hours of the experiment did not only occur for *pchEF* (Figs. 3e, S6e), but also for *pvdA* (Figs. 3f, S6f). Moreover, the bimodality in *pchEF* expression persisted at many of the later time points, especially under conditions with low apo-transferrin concentration (Figs. 3e, S6e).

Taken together, the single-cell analysis reveals a funnelling pattern in the expression of siderophore synthesis genes in all iron-limited media, from high between-cell heterogeneity (including bimodality) during the early time points to a more aligned homogenous expression patterns during later time points.

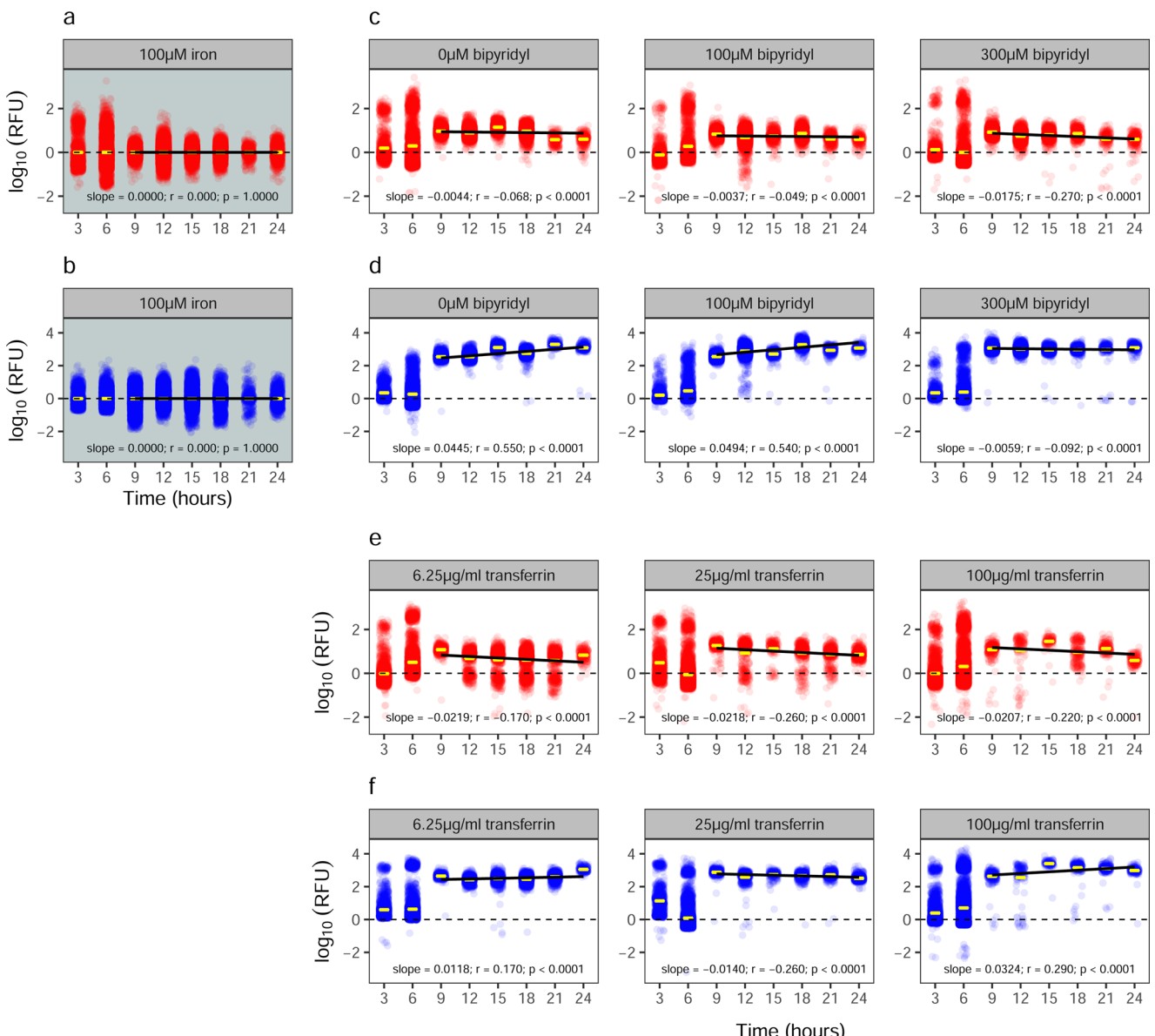

**Fig. 3 Single-cell siderophore gene expression patterns across time and media.** The expression of pyochelin (red: *pchEF*) and pyoverdine (blue: *pvdA*) synthesis genes measured with the double reporter PAO1*pvdA*::*mcherry*–*pchEF*::*egfp* across a range of CAA media differing in their levels of iron limitations. Each dot represents an individual cell with gene expression shown as relative fluorescent unit (RFU), representing the blank-corrected log-transformed integrated fluorescence density, log(IntDen). Values > 0 represent cells with gene expression above background level. **a**, **b** Iron-replete CAA medium (100 μM FeCl₃); **c**, **d** CAA media with increasing concentration of the iron chelator bipyridyl; **e**, **f** CAA media with increasing concentration of the iron chelator apo-transferrin. Yellow bars represent the mean gene expression of all cells at a specific timepoint. Black solid lines indicate correlation trend lines of gene expression from the 9th hour onwards. The Pearson correlation coefficient *r* and the *p*-value are provided in each panel together with the slope of the trendline.

Moreover, there was a temporal decline in pyochelin gene expression from the 9th hour onwards, while pyoverdine gene expression remained steady or even increased during the same time period. Given the high heterogeneity among cells, the strength of the correlations is moderate, yet they confirm the observations made at the batch-culture level (Fig. 2) that there is a temporal relative transition towards lower pyochelin and higher pyoverdine gene expression over time.

**Positive correlations between pyochelin and pyoverdine gene expression across cells.** We then tested for correlations in the expression of pyochelin and pyoverdine synthesis genes across cells. Positive correlations would indicate that individuals in a

clonal population segregate along a continuum from low to high siderophore producers, while negative correlations would suggest that cells specialize in either pyochelin or pyoverdine production. We observed positive correlations between pyochelin and pyoverdine gene expression for all time points and media conditions (Figs. 5a, b and S8), refuting the specialization but supporting the continuum hypothesis.

Under the iron-supplemented condition, the positive correlations were initially strong, but then significantly declined over time ($F_{1,6} = 46.57$, $p = 0.0005$). In CAA medium with bipyridyl, the correlation coefficients also significantly declined over time (ANCOVA: $F_{1,18} = 34.66$, $p < 0.0001$), with the decline being steeper under less stringent iron limitation (significant interaction

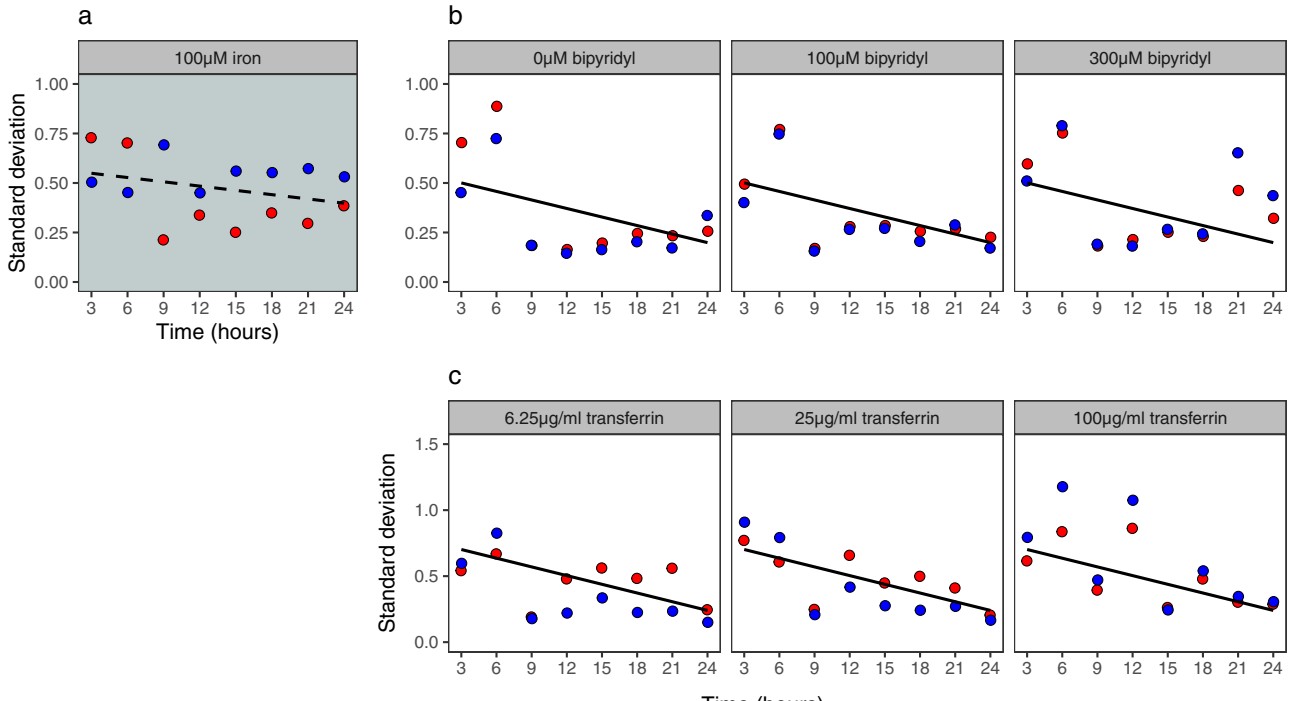

**Fig. 4 Heterogeneity in siderophore gene expression declines over time.** Temporal heterogeneity in siderophore gene expression for pyochelin (red: *pchEF*) and pyoverdine (blue: *pvdA*), measured with the double reporter strain PAO1*pvdA*::*mcherry–pchEF*::*egfp* across a range of CAA media differing in their levels of iron limitations. Heterogeneity in gene expression is shown as the standard deviation across log-transformed fluorescence values of all cells. **a** Iron-replete CAA medium (100 μM FeCl₃); **b** CAA media with increasing concentration of the iron chelator bipyridyl; **c** CAA media with increasing concentration of the iron chelator apo-transferrin. Solid black trendlines indicate significant declines in heterogeneity across time.

between time and media: $F_{1,18} = 5.423$, $p = 0.0143$). In CAA medium supplemented with apo-transferrin, the correlation coefficient also significantly declined over time ($F_{1,20} = 5.061$, $p = 0.0359$), and similarly so for all conditions ($F_{2,20} = 0.326$, $p = 0.7254$).

**Positive correlations between siderophore and housekeeping gene expression across cells.** To better understand the positive correlations, we examined whether they arise because cells in clonal populations differ in their overall metabolic state, and whether the more active cells are the ones that show higher investments into siderophores. Here, we do not assume any specific mechanism being in place, but simply argue that cells with low metabolic activity would express siderophore genes (and any other gene) at a reduced rate. To test this idea, we simultaneously quantified the expression of one of the siderophore genes (*pchEF* or *pvdA*) together with the *rpsL* housekeeping gene within the same cell.

In this context, we first tested whether *rpsL* gene expression is a good proxy for the overall metabolic activity of a cell. For this purpose, we subjected the PAO1*rpsL:mcherry* reporter strain to a concentration gradient of trimethoprim, an antibacterial compound known to inhibit metabolism[50–52]. We found that trimethoprim significantly reduced overall bacterial growth and *rpsL* expression levels in a concentration-dependent manner (Fig. S9). More importantly, trimethoprim reduced the per capita *rpsL* expression levels especially at earlier time points (up to 12 h, Fig. S9g), showing that *rpsL* gene expression correlates with metabolic activity of cells.

We then analyzed the relationship between the expression of *rpsL* and the two siderophore genes. We observed positive correlations for both *pchEF* and *pvdA*, for all time points and media conditions (Figs. 5c, d and S10a, b). These results strongly

suggest that the overall metabolic state of a cell dictates its siderophore investment levels. However, we also observed that the correlation coefficients were significantly weaker for the *pchEF-rpsL* than for the *pchEF-pvdA* gene pair (paired t-test: $t_{11} = 6.70$, $p < 0.0001$), while there was no difference in the strength of the correlation between the *pvdA-rpsL* and *pchEF-pvdA* gene pairs ($t_{11} = 1.14$, $p = 0.2780$). These findings indicate that investment into pyoverdine depends more stringently on the metabolic state of the cell than for pyochelin.

## Discussion

The aim of our study was to obtain a detailed view on siderophore gene expression at the single cell level in growing populations of the bacterium *Pseudomonas aeruginosa*. This species produces two siderophores, pyochelin and pyoverdine, and the regulatory mechanisms governing the expression of these secondary metabolites are well known[22]. However, unknown is whether the seemingly fine-tuned regulation of siderophores, in response to iron availability and cell density, observed at the population level is driven by coordinated homogenous behaviours of the individual cells in the population. Our null-hypothesis was that there is no coordination such that the population-level response simply reflects the sum of its noisy individuals[17]. We could refute this hypothesis, as we observed that cells switched from initially heterogenous to highly homogenized gene expression patterns over the population growth cycle. One of our alternative hypotheses was that coordination could promote specialization, whereby subpopulations of cells either invest in pyoverdine or pyochelin[45]. We could also refute this hypothesis, as we found no evidence for negative correlations in the expression of the two traits across cells. Instead, our data reveal a funnelling pattern, where cells switch from an initially uncoordinated highly noisy siderophore gene expression phase to a state, where cells align

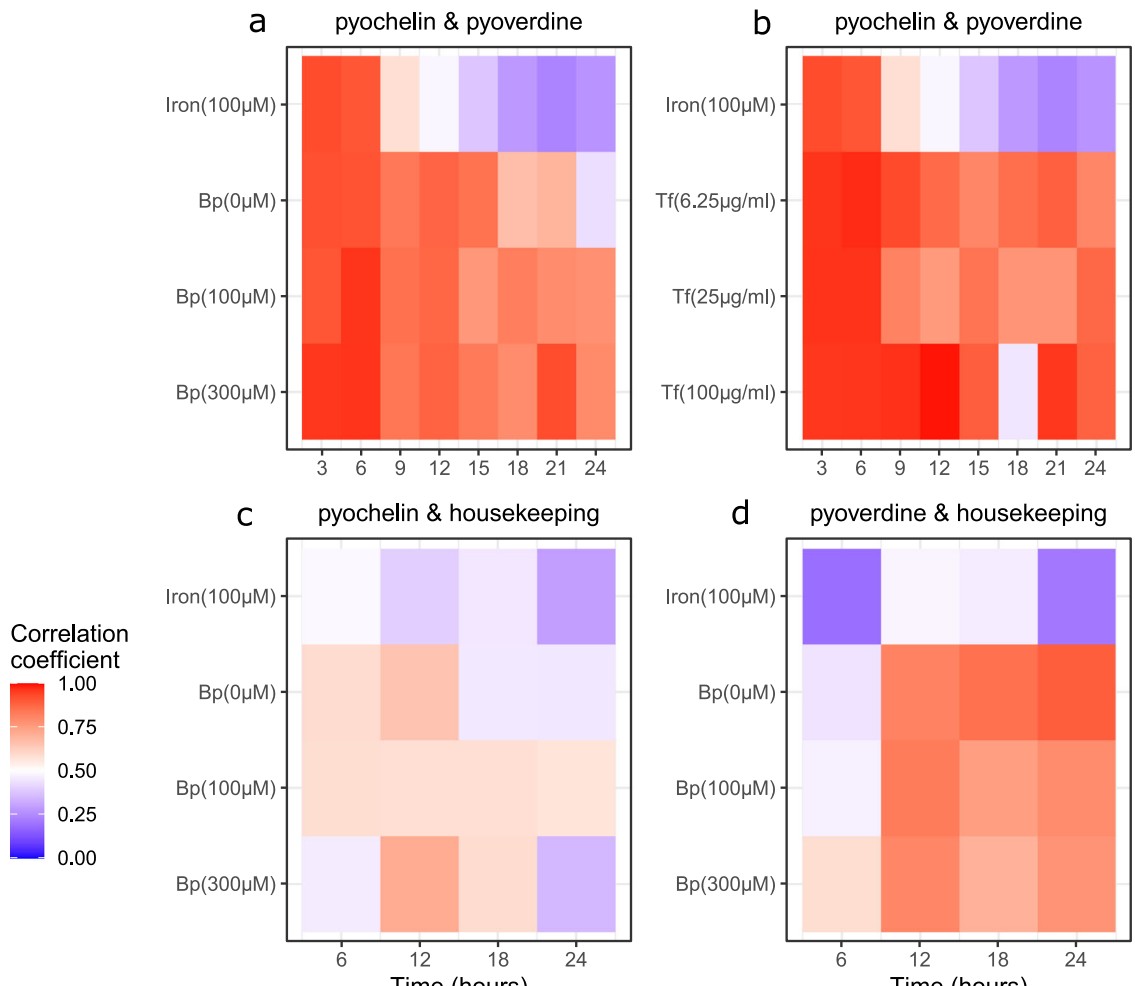

**Fig. 5 Correlations between pyochelin, pyoverdine, and housekeeping gene expression among clonal cells across time and media.** Correlation coefficients are shown as heatmaps, and indicate the strength of associations in the expression of two genes across individual cells within a population. **a**, **b** Correlations between pyochelin (*pchEF*) and pyoverdine (*pvdA*) gene expression, measured with the double reporter strain PAO1*pvdA::mcherry–pchEF::egfp* in iron-supplemented CAA medium and in CAA media supplemented with either increasing concentrations of the iron chelators bipyridyl or apo-transferrin. **c** Correlations between pyochelin *pchEF* and housekeeping *rpsL* gene expression, measured with the double reporter PAO1*pchEF::mcherry–rpsL::egfp* in iron-supplemented CAA medium and in CAA media supplemented with increasing concentrations of the iron chelator bipyridyl. **d** Correlations between pyoverdine *pvdA* and housekeeping *rpsL* gene expression measured with the double reporter PAO1*pvdA::mcherry–rpsL::egfp* in iron-supplemented CAA medium and in CAA media supplemented with increasing concentrations of the iron chelator bipyridyl.

their siderophore investments to similar levels. Concomitant with this alignment, there was a moderate but significant relative transition towards lower pyochelin and higher pyoverdine investment levels over time. This temporally stratified gene expression pattern is consistent across different iron-limited environments and is reminiscent of chronobiological (cyclical physiological) processes known from multi-cellular organisms[53,54].

We propose that the chronobiological cycle of siderophore gene expression entails three different phases (I to III), covering the time spans from low to high population density, that are steered by the various interconnected regulatory mechanisms governing siderophore synthesis (Fig. 6). Phase I comprises the first six hours of our experiment, during which both pyochelin and pyoverdine gene expression is highly heterogenous across cells. Heterogeneity is often characterized by bimodal gene expression, where the majority of cells remains in the off-stage, while a minority of cells show high (often overshooting for pyochelin) siderophore gene expression. One conceivable

possibility is that this bimodality is driven by differences in the internal iron stocks of cells (Fig. 6a). *P. aeruginosa* uses bacterioferritin, a hollow protein that can take up several thousands of iron ions, for intra-cellular storage[55,56]. Since our cells come from iron-rich overnight LB cultures, we can assume that their storages are replenished but that there is between-cell variation in iron stocks (e.g., younger cells might have lower stocks than older cells). While the stocks allow initial survival and growth after the transfer to iron-limited media[57], Fur-mediated repression should be released and siderophore gene expression start upon stock depletion[20,23–25]. Since each cell responds individually to its internal iron stocks, bimodality can most parsimoniously be explained by inter-individual variation in iron stocks.

Phase II involves the timeframe from 6 up to 15 h, during which heterogeneity in siderophore gene expression is greatly reduced, and all cells switch to an on-state. Phase II coincides with the exponential growth phase of populations (Fig. S3). We propose that siderophore-mediated signalling of cells triggers the homogenization of gene expression (Fig. 6b). As mentioned in the

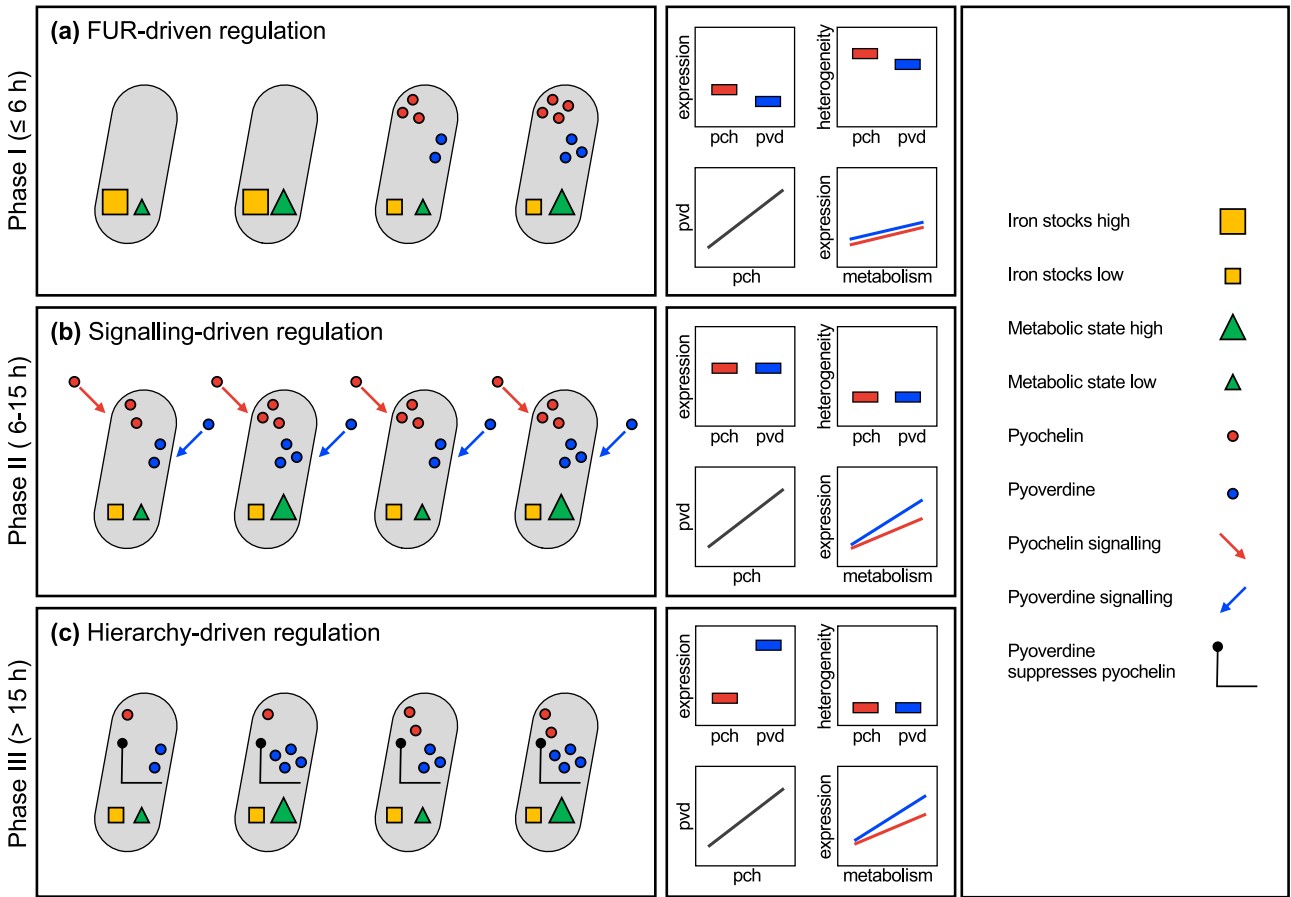

**Fig. 6 A three-phase model of how regulatory elements, internal iron stocks, and the metabolic state of cells can explain decision-making processes and the chronobiological cycle of siderophore gene expression in growing populations of *P. aeruginosa*. a** Phase I (up to 6 h) is driven by Fur (ferric uptake regulator) and variation in internal iron-stocks across cells. Cells with low stocks start to express and produce pyochelin and pyoverdine independent of their metabolic state. This induces high gene expression heterogeneity (including bimodality) across cells and strong positive correlations between pyochelin and pyoverdine expression. **b** Phase II (between 6 and 15 h) is driven by signalling-mediated regulation and variation in the metabolic state. Siderophore-mediated signalling leads to homogenized and increased levels of gene expression. Residual variation across cells is largely explained by metabolic state differences. **c** Phase III (>15 h) is driven by hierarchy-driven regulation, inducing a shift from pyochelin to pyoverdine gene expression. Residual variation across cells is still explained by metabolic state differences, and more so for pyoverdine than for pyochelin.

introduction, signalling helps to fine-tune siderophore production and occurs when ferri-pyoverdine binds to the membrane-bound cognate receptor FpvA and through the import of ferri-pyochelin via FptA that acts as a co-factor for PchR[27,28,30,31]. Signalling is likely stochastic at low population density and low siderophore concentrations as experienced in Phase I. Meanwhile, signalling is likely more deterministic in Phase II, where siderophore concentrations and cell density increases, and the signals can be shared more reliably between cells, leading to the homogenization of behaviours across cells[14]. The increased sharing and uptake of pyoverdine is evidenced by our single-cell data, showing that the correlation between pyoverdine gene expression and molecule fluorescence becomes weak during this time period (Fig. S5).

Phase III involves the time points from 15 h onwards, where cells continue growing (Fig. S3). Here, we observed continued low gene expression heterogeneity across cell and a moderate but significant transition in the relative gene expression from pyochelin to pyoverdine. While the low heterogeneity is likely the result of ongoing siderophore-signalling, the shift towards reduced pyochelin gene expression could be guided by the hierarchical regulatory link between the siderophores, where high pyoverdine availability inhibits pyochelin synthesis[26,32] (Fig. 6c). As mentioned in the introduction, the transition could be induced by the increasing quantities of the high-iron affinity pyoverdine in

the media that bind away the iron and thereby inhibit pyochelin-mediated signalling and synthesis. Taken together, the integration of three regulatory elements (Fur, siderophore-signalling and hierarchy) could guide the chronobiological cycle of siderophore gene expression in populations of *P. aeruginosa*, a pattern that is expected to be reinitiated in each new growth cycle starting from low cell density.

Despite the fact that siderophore gene expression became homogenized across cells over time, there was still considerable inter-individual variation (Fig. 3). One aspect that could contribute to this variation is the fact that siderophore synthesis is influenced by additional regulatory elements than the ones focussed on here (Fig. 6). For example, transcription factor network analyses revealed complex interactions between PvdS and multiple other regulatory elements[39,41]. All these regulatory links are likely noisy and can contribute to inter-individual heterogeneity in siderophore gene expression. Moreover, our analyses suggest that inter-individual variation could be associated with differences in the metabolic state of cells. Our reasoning is based on the observation that the expression of both siderophore genes (*pchEF* and *pvdA*) correlated positively with the *rpsL* housekeeping gene expression (Fig. 5). The *rpsL* gene encodes the 30S ribosomal protein S12, which plays a major role in translational accuracy. While *rpsL* is essential and expressed by all cells, the

expression magnitude can be taken as a proxy for metabolic activity[58]. We confirmed this notion by showing that the addition of the metabolic inhibitor trimethoprim decreased the per capita *rpsL* gene expression (Fig. S9g). Interestingly, we observed that the strength of the positive correlations between *rpsL* and siderophore genes varied across the phases I–III and also differed between the two siderophores (Figs. 5 and 6). In phase I, the association between *rpsL* and siderophore gene expression was relatively weak, indicating siderophore gene expression is poorly linked to the metabolic state of cells (Fig. 6a). In contrast, stronger correlations between metabolic activity and siderophore gene expression arose in phase II and III. Since these phases cover the exponential growth period during which cells are metabolically most active, our data suggest that cells reaching a higher metabolic state make more siderophores. Finally, we found that the expression of pyoverdine was more strongly correlated with metabolic activity than pyochelin expression (Fig. 5c, d). This weaker association could arise because the small pyochelin is cheaper to produce than the large pyoverdine, meaning that the production of pyochelin is less dependent on the metabolic state of a cell[26].

While the above sections aim to link siderophore gene expression patterns to population growth stages, regulatory mechanisms, and metabolic activity, we here offer possible adaptive (evolutionary) explanations for the observed patterns. An obvious explanation is that the specific regulatory circuits have evolved to optimize the cost-to-benefit ratio of siderophore investment under iron-limited conditions[26,43]. However, there could be more to it. For example, our data suggest that the role of pyochelin might have been underestimated in the past[32]. While typically considered as a secondary siderophore, we show that pyochelin gene expression occurs under all conditions and peaks early during the growth cycle relative to pyoverdine gene expression (Fig. 3). One adaptive explanation could be that bacteria follow a two-step iron-scavenging strategy when colonizing a new patch. They first invest in the cheap pyochelin during the early colonization phase, where population density is low[5]. They then switch to the more expensive pyoverdine when reaching higher population densities, where the sharing of siderophores becomes more reliable. The 'early-on' scavenging role of pyochelin is further supported by our observation that there is always a fraction of cells in the population that have pyochelin expression switched on, even under iron-rich conditions. This could point towards a bet-hedging strategy[18,59,60], where clonal populations maintain a fraction of cells in a pyochelin-on state, in order to be immediately able to react to an environmental change in iron limitation. This strategy could provide a competitive edge over a strategy, where all cells are in a siderophore-off stage, because it cuts short the initial sensing of iron limitation and the mounting of siderophore synthesis from scratch[61].

In summary, our single-cell gene expression study reveals a transition from initially noisy to coordinated and aligned siderophore gene expression patterns in clonal populations of *P. aeruginosa*. Our data indicate that clonal bacterial populations can coordinate their actions in ways that match decision-making processes observed in higher organisms[62–65]. Specifically, decision-making theory predicts that individuals can coordinate their actions when integrating information from their local neighbours. The theory further predicts that the response accuracy can be increased when information is sampled across longer time periods and/or a broader range (e.g., across more individuals)[14,66]. We postulate that siderophore signalling could fulfil the function of information collection, whereby the aggregate availability of siderophores at the local group level would be the currency of information. It provides information on cell density and the siderophore investment of neighbours. The accuracy of information collection increases with cell density and could explain why the aligned, more homogenous response only emerges after some time. Important to note is that collective decision-making does not require cognitive abilities and thus could be based on simple feedback mechanisms like siderophore signalling. Overall, our single-cell approach offers a deeper understanding on bacterial social behaviours among cells in clonal groups.

## Methods

**Strains and strain construction**. We used the standard laboratory strain *P. aeruginosa* PAO1 (ATCC 15692) for all our experiments, which produces the siderophores pyoverdine and pyochelin[26,67]. In the genetic background of this wild type strain, we chromosomally integrated fluorescent gene reporter constructs to quantify siderophore gene expression. All constructs were integrated at the *att*Tn7 site using the mini-Tn7 system[68]. We used the following single-gene reporter fusions previously constructed in our laboratory[69]: PAO1*pvdA:mcherry*, PAO1*pchEF:mcherry*, and PAO1*rpsL:mcherry*, in which the promoter of the pyoverdine biosynthetic gene *pvdA*, the pyochelin biosynthetic genes *pchE* and *pchF*, and the housekeeping gene *rpsL* are fused to *mcherry*, respectively.

To simultaneously track the expression of two genes within an individual, we constructed double gene expression reporters. We constructed three different reporter strains involving the following pairs of genes: (1) PAO1*pvdA::mcherry–pchEF::egfp*, (2) PAO1*pchEF::mcherry–rpsL::egfp*, (3) PAO1*pvdA::mcherry–rpsL::egfp*. The genetic scaffold of the double reporter strain (1) is based on the construct proposed by Minoia et al.[70], where the two promoter fusions are located on the opposite strands, to minimize promoter interference (Fig. S1). For the double reporter strains (2) and (3), we developed an improved scaffold, where the promoter fusions are sequentially arranged on the leading strand (Fig. S2). Promoter interference was eliminated through the insertion of multiple terminator sites between the two promoter fusions. The bacterial strains, plasmids, and primers used in this study are listed in the Supplementary Tables S2–S4, respectively. Important to note is that all double reporters were inserted as single-copies at the attTn7 site (as described above), which distinguishes our system from a previously reported plasmid-based multi-reporter system[71].

**Growth conditions**. Prior to experiments, overnight cultures were grown in 8 ml Lysogeny broth (LB) in 50 ml tubes, incubated at 37 °C, 220 rpm for approximately 18 h. Cells were harvested by centrifugation (8000 rpm for 2 min), subsequently washed in 0.8% saline and adjusted to OD600 = 1 (optical density at 600 nm). For all batch culture and microscopy experiments, the harvested cells were grown in CAA medium (5 g casamino acids, 1.18 g $K_2HPO_4*3H_2O$, 0.25 g $MgSO_4*7H_2O$, per liter), buffered at physiological pH by the addition of 25 mM HEPES. The initial dilution of bacteria in CAA medium was OD600 = 0.0001. To induce iron limitation, we supplemented CAA with one of two iron chelators: the synthetic 2–2′-bipyridyl or the natural human apo-transferrin. While both iron chelators bind the extracellular iron, bipyridyl is also cell-permeable, and can be mildly toxic and interfere with intra-cellular iron homeostasis[72–74]. The concentration ranges used were 50–300 μM for bipyridyl and 6.25–100 μg/ml for apo-transferrin. In the case of apo-transferrin, we also supplemented CAA medium with 20 mM $NaHCO_3$, an essential co-factor[75]. We further created an iron-replete condition by supplementing CAA with 100 μM $FeCl_3$. All chemicals were purchased from Sigma Aldrich (Buchs SG, Switzerland).

**Population-level growth, siderophore gene expression, and pyoverdine production**. We first quantified growth and siderophore gene expression of *P. aeruginosa* at the population level, by growing bacteria in CAA medium across a gradient of iron limitation (seven different conditions each for the experiments with either bipyridyl or apo-transferrin). In addition, we also measured pyoverdine production by tracking the natural fluorescence of this molecule. We used the wildtype PAO1, and the single gene expression reporters PAO1*pvdA:mcherry* and PAO1*pchEF:mcherry*. Experiments were carried out in 200 μl of medium distributed on a 96-well plate. Bacteria were inoculated at a starting density of OD600 = 0.0001 and incubated at 37 °C for 24 h in a multimode plate reader (Tecan, Männedorf, Switzerland). Cultures were shaken every 15 min for 15 s prior to measuring growth (OD600), mCherry fluorescence (excitation:582 nm/emission:620 nm) and pyoverdine fluorescence (excitation:400 nm/emission:460 nm). We applied blank subtractions and standardized the relative fluorescence units (RFU) by OD600 (RFU/OD600). Each experiment featured four replicates per growth condition and was repeated twice.

We found that the growth integral (area under the curve) over 24 h did not significantly differ between the wildtype PAO1 and the two single gene reporter strains (for bipyridyl: $F_{5,162} = 0.0158$, $p = 0.9843$; for transferrin: $F_{5,162} = 0.1357$, $p = 0.8732$). We therefore combined data from all three strains for growth analyses (Figs. 1, S3).

**Population level growth and *rpsL* gene expression with trimethoprim.** Trimethoprim is a metabolic inhibitor that interferes with the bacterial enzyme that converts folic acid into tetrahydro-folic acid, which is a key co-enzyme in the synthesis of nucleic acids[50–52]. It is used here to test whether the expression of the housekeeping gene *rpsL* responds to trimethoprim and thus can be taken as a proxy for metabolic activity of cells. We exposed PAO1*rpsL:mcherry* to a range of trimethoprim concentrations (0, 0.5, 1, 2, 5, 10, and 20 µg/ml) in CAA medium buffered at physiological pH by the addition of 25 mM HEPES, in three-fold replication. The experiment was carried out in 200 µl of medium distributed on a 96-well plate. Bacteria were inoculated at a starting density of OD600 = 0.0001, and incubated at 37 °C for 24 h in a multimode plate reader (Tecan, Männedorf, Switzerland). Cultures were shaken every 15 min for 15 s prior to measuring growth (OD600) and mCherry fluorescence (excitation:582 nm/emission:620 nm). To obtain dose-response curves, we calculated the growth and gene expression integral over 24 h, and scaled all values relative to the 0 µg/ml trimethoprim treatment. We further calculated the per capita *rpsL* expression by dividing mCherry fluorescence values by OD600, and tested whether this ratio declines with higher trimethoprim concentrations, which would indicate that *rpsL* gene expression is a good proxy for metabolic activity of cells and cultures.

**Siderophore gene expression at the single-cell level**. We grew bacteria in batch cultures across a range of iron limitations in CAA (see above) in 1.5 ml volumes distributed on 24-well plates, using a starting inoculum of OD600 = 0.0001. The cultures were incubated under shaken condition (170 rpm) for 24 h at 37 °C. From these growing cultures, we took small aliquots of 2 µl every 3 h (upto 24 h), which we put on a microscopy slide for the quantification of single-cell gene expression using fluorescence microscopy. Because of the heavy workload associated with our 3 h imaging interval across 24 h, we split each experiment into two time blocks. We set up a first plate with experimental conditions from overnight LB cultures early in the morning of day 1 and measured gene expression of cells at times 3rd, 6th, and 9th hour on day 1 and at the 21st and 24th hour on day 2. In addition, we transferred the overnight culture to fresh LB medium in the morning of day 1, to keep a growing bacterial population. This culture was then used to set up a second plate with experimental conditions in the evening of day 1. We used this second plate to measure gene expression of cells at the 12th, 15th, 18th, and 21st hours on day 2.

We performed our single-cell experiments with a subset of growth conditions used for our population-level experiments. Specifically, we used CAA medium supplemented with 100 µM FeCl₃ as the iron-replete condition, and gradually increased iron limitation by either adding no iron, 100 µM, or 300 µM of bipyridyl. We further repeated the experiments by adding either 6.25, 25, or 100 µg/ml of apo-transferrin.

We had two main experimental blocks. In block 1, we examined pyochelin and pyoverdine gene expression (separately for bipyridyl and apo-transferrin). Each replicate within block 1 featured the following four strains: (1) PAO1*pvdA:mcherry* single reporter; (2) PAO1*pchEF:mcherry* single reporter; (3) PAO1*pvdA::mcherry–pchEF::egfp* double reporter; and (4) the wildtype PAO1. In block 2, we compared siderophore with house-keeping gene *rpsL* expression. Each replicate within block 2 featured the following four strains: (1) PAO1*pvdA::mcherry–rpsl::egfp*; (2) PAO1*pchEF::mcherry–rpsl::egfp*; (3) PAO1*pvdA::mcherry–rpsl::egfp*; and (4) the wildtype PAO1. Each experimental block featured two full replicates including all conditions and time points. Block 1 was repeated another two times including the early time points only to compensate for the fact that cell number was low during the early growth phase.

**Preparation of microscope slides**. For microscopy, we adapted a method previously described by de Jong et al.[76] and Weigert and Kümmerli[49]. Standard microscope slides (76 mm × 26 mm) were sterilized with 70% ethanol. We used 'Gene Frames' (Thermo Fisher Scientific, Vernier, Switzerland) to prepare agarose pads on which bacteria were seeded. Each frame features a single chamber (17 mm × 28 mm) of 25 mm thickness. The frames are coated with adhesives on both sides so that they stick to the microscope slide and the coverslip. The sealed chamber is airproof, which prevents pad deformation and evaporation during experimentation.

To prepare agarose pads, we heated 50 ml of 25 mM HEPES buffer with agarose (1%) in a microwave. The agarose-buffer solution was cooled to approximately 50 °C. We pipetted 700 µl of the solution into the gene frame and immediately covered it with a sterile coverslip. The coverslip was gently pressed to let superfluous medium escape. Slides were stored overnight at 4 °C to ensure pad solidification. We removed the coverslip (by carefully sliding it sideways) and divided the agarose pad into six smaller pads of roughly equal size with a sterile scalpel. We introduced channels between pads, which served as oxygen reservoirs. Every pad received 2 µl of a growing bacterial culture (of a specific time point, strain, and experimental condition) extracted from the 24 well plates. We diluted cells in 0.8% saline to get an optimal number of individually discernible cells. The dilutions ranged from 1:1 at 3 h to 1:100 at 24 h. Upon the addition of bacteria, we let the agarose pads air-dry for 2 min, and then sealed them with a new sterile coverslip.

**Microscope set-up and imaging**. We immediately imaged the bacteria on the pads at the Centre for Microscopy and Image Analysis of the University of Zurich (ZMB) using an inverted widefield Olympus ScanR HCS microscope featuring the OLYMPUS cellSens Dimensions software. Images were captured with a PLAPON 60× phase oil immersion objective (NA = 1.42, WD = 0.15 mm) and a Hamamatsu ORCA_-FLASH 4.0V2, high sensitive digital monochrome scientific cooled sCMOS camera (resolution: 2048 × 2048 pixels, 16-bit). For fluorescence imaging, we used a fast emission filter wheel, featuring a FITC SEM filter for eGFP (excitation = 470 ± 24 nm, emission = 515 ± 30 nm, DM = 485), a TRITC SEM filter for mCherry (excitation = 550 ± 15, emission = 595 ± 40, DM = 558) and a DAPI SEM filter for pyoverdine auto-fluorescence (excitation = 395 ± 25, emission = 435 ± 26, DM = 400). We imaged at least six fields of view per pad, with each pad representing a specific combination of bacterial strain, experimental condition, and time point.

**Image processing**. For image processing, we established a semi-automated workflow[49]. The workflow starts with the machine learning, supervised object classification, and segmentation tool ILASTIK[77]. It features a self-learning algorithm that classifies objects (cells in our case) from the background using phase-contrast images. We used around 20 representative phase-contrast images from our experiments to train ILASTIK. After training, we supervised the results, marked errors and, re-initiated the next training round, until segmentation was optimized and nearly error-free. We then used the trained algorithm to segment all our images in a fully automated process[49]. Subsequently, we processed the segmented phase-contrast images with the open-source software FIJI[78], where regions of interest (ROI) were defined, capturing cells. We extracted information on cell size and fluorescence intensity for every single cell. To exclude segmented artifacts like cell debris, we set a cut-off area for ROIs, below which segmented objects were excluded.

We applied three different correction steps to our fluorescence images. While all three steps were carried out independently for the green (eGFP) and red (mCherry) fluorescence channels, only the first two steps were carried out for blue (pyoverdine) fluorescence channel. First, we corrected for agarose pad autofluorescence. For each agarose pad, we imaged at least 4 empty random positions without bacteria and averaged the grey values across pixels in the respective fluorescent channel. This average grey value was then subtracted from the fluorescence images containing cells. Then we overlaid the ROIs obtained from the phase-contrast images with the corresponding fluorescence images. Second, we corrected for intensity difference across the fields of view caused by microscope vignetting. To achieve this, we quantified the average grey value of the area outside the ROIs of each image containing cells, and subtracted that average grey value from all ROIs in that particular image. At the end of this step, we obtained background-corrected fluorescence values for each individual cell (i.e., ROI). We used the Integrated Density (IntDen) values, which is the mean grey value multiplied by the area of the cell, for further analysis. In the last step, the IntDen were corrected for the auto-fluorescence of bacterial cells. Specifically, we measured the fluorescence of PAO1 wildtype cells that did not have a fluorescence reporter. Since the wildtype strain always ran in parallel with the reporter strains, we obtained measures of auto-fluorescence as IntDen values for a large number of wildtype cells for each time point and experimental condition. We log-transformed the IntDen values of wildtype cells and calculated the log(IntDen) median fluorescence value across all wildtype cells per time point and condition. We then subtracted these log(IntDen) median values from all individual cells of the respective time point and condition. A consequence of this correction procedure is that the log(IntDen) median value for the wildtype equals zero for all time points and conditions, and log(IntDen) values > 0 indicate gene of interest is expressed. We did not correct for the auto-fluorescence of bacterial cells in the blue (pyoverdine) fluorescence channel because the wildtype also produces pyoverdine like the reporter strains. Using this procedure, we quantified gene expression of 327113 cells.

**Statistics and reproducibility**. Each experimental block was repeated two to four times. While the results between repeats were reproducible in terms of relative gene expression patterns, the absolute level of gene expression naturally varied between repeats. For quantitative data analysis, we aimed to fuse the data sets from the individual repeats. This required a data normalization step. First, we calculated the mean gene expression $x_r$ for every condition, time point, and strain within each repeat. Next, we calculated the global mean $\bar{x}$ across repeats for the condition, time point, and strain in question. Then, we calculated the difference between the two means $\triangle x = \bar{x} - x_r$. Finally, we subtracted $\triangle x$ from every single cell from the corresponding condition, time point, and strain.

We used general linear models for statistical data analysis in R 3.4.2. For our population-level growth and gene expression analysis, we used analysis of variance (ANOVA) models with strain type as fixed factor and iron chelator concentration as co-variate. We conducted separate analyses for the two iron chelators bipyridyl and apo-transferrin. We used linear regression analysis to test whether siderophore gene expression changes over time after it is induced. For this analysis, we considered all time points starting from the 9th hour, the time point at which all cells show siderophore gene expression. Next, we compared the standard deviation in gene expression across cells, as a measure of heterogeneity, across time and media conditions. We used ANOVA models, where we fitted media condition as a

fixed factor and time as a covariate. Finally, we calculated Pearson's correlation coefficient $r$ to relate the expression of two genes across cells for any given condition tested. We then extracted all $r$-values and used ANOVAs to test whether the strength of correlations varies as a function of media condition (fixed factor) and time (co-variate). Finally, we conducted paired $t$-tests to compare whether correlation coefficients differ between the expression of the two siderophore genes (*pchEF* vs. *pvdA*), and the expression of a siderophore gene (*pchEF* or *pvdA*) and the housekeeping gene *rpsL*. We log-transformed all single-cell gene expression values prior to analysis.

**Reporting summary**. Further information on research design is available in the Nature Research Reporting Summary linked to this article.

## Data availability

The datasets generated during and/or analyzed during the current study are available in the Figshare repository[79].

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

## Acknowledgements

We thank Priyanikha Jayakumar, Chiara Rezzoagli, Tobias Wechsler, Jos Kramer, and Michael Weigert for help in the lab and with data analysis, and the Centre for Microscopy and Image Analysis for support. This project has received funding from the European Research Council (ERC) under the European Union's Horizon 2020 research and innovation programme (grant agreement no. 681295).

## Author contributions

S.M. and R.K. designed the study. S.M. carried out all experiments and constructed some of the strains. S.M. and R.K. analyzed the data and wrote the paper.

## Competing interests

The authors declare no competing interests.
