## [Peer Review File · Communications Biology]

Reviewers' comments:

Reviewer #1 (Remarks to the Author):

The aim of this study was to get an insight into the cross-talk and interactions between the two siderophores produced by *Pseudomonas aeruginosa*, pyochelin (PCH, low affinity for iron) and pyoverdine (PVD, high affinity for iron). This work is a follow-up of the nice paper of the same group (Dumas et al. 2013) who showed for the first time that PCH production precedes the PVD production once iron limitation is increasing, but this study was done on batch population, and not at individual cell level, which is the case here. They used to this effect reporter genes PAO1pchEF-mcherry and pvdA mcherry and showed that in batch cultures in CAA iron poor medium the production of PVD increases with the degree of Fe limitation caused by increasing concentrations of dipyriddy or by transferrin. PCH production quickly plateaued while PVD increases, confirming previous data. At single cell level (an enormous work), using a double reporter (pvdA mcherry-pchEF egfp) they observed first a dual population of PCH producing/non producing cells before the exponential phase, this bimodal expression disappearing during the growth under increasing iron limitation conditions imposed by the presence of transferrin or bipyridyl (figures 3 and 4). They also observed a positive correlation between PCH and PVD gene expression across cells and also between the siderophore genes expression and the expression of the housekeeping rpsL gene, this correlation being weaker for pchEF-rpsL than for pchEF-pvdA. Finally, they present a model explaining their observations where different factors influence the production of the two siderophores, Fur regulation, and the metabolic activity (phase I). In phase II the siderophore-mediated cell signalling regulation is taken into account and in phase III the PVD-mediated PCH repression.

While I agree with the model and find the study well done and quite interesting, still some questions arise, which should be addressed in the discussion.

- First of all, the problem of PVD-mediated PCH production repression. While it makes sense, there is apparently no explanation for it. One could be that the increased PVD deprives iron from PCH and hence impairs the Fe-PCH binding to the PchR regulator and at the same time causes a decreased availability of the Fe-PCH receptor FptA. Some information can eventually be found in the articles from Michel et al. from the Lausanne group. There is also an old paper from Poole showing that PchR can be a repressor and an activator (like many AraC). But I agree that it is speculative since not much is known about the PchR regulon. It is interesting also to mention that the secondary siderophore quinolobactin of *P. fluorescens* ATCC 17400 production is also repressed by PVD (the QB regulator is also an AraC).

- Line 221: It is true that we know how PCH and PVD production is regulated (Fur, PchR for PCH), Fur-PvdS/FpvI for PVD and its receptor, respectively), but it is more complex than that since in the recent years other regulators have been shown to be involved in the complex regulation of PVD biosynthesis and uptake (OxyR for pvdS) (Qing et al. 2013), other sigma factors are involved as well, such as SigX (Schulz et al., 2015). Very recently there is an interesting article published in ELife where the authors used a Selex approach to determine the targets of all transcription factors (not sigmas) in *P. aeruginosa* and they present in their supplementary figure 9 that 14 regulators are involved in iron uptake regulation (8 for pvdS! while SouR AraC regulator could be involved in the regulation of fpvB (second PVD receptor) and pchR regulation, somehow linking the two siderophores. I just mention this to show that the regulation is quite more complex than expected but fact does not contradict the data obtained in this study since the metabolic activity is likely to be influenced by a plethora of regulators.

Reviewer #2 (Remarks to the Author):

In the Subham Mridha and Rolf Kümmerli paper, the authors followed indirectly the transcription and expression levels of the enzymes involved in the biosynthesis of the siderophores pyoverdine and pyochelin in *P. aeruginosa*. Based on the data obtained concerning these genes transcription

and expression, they propose that the chronobiological cycle of siderophore gene expression entails three different phases. The first six hours, both pyochelin and pyoverdine gene expression are highly heterogeneous across cells (the majority of cells remains in the off-stage, while a minority of cells shows high siderophore gene expression, mostly pyochelin). In phase 2, heterogeneity in siderophore gene expression is greatly reduced, and all cells switch to an on-state and it coincides with the exponential growth phase. Phase 3 corresponds to a shift in the relative gene expression from pyochelin to pyoverdine, with low heterogeneity across cells. The paper is nicely written and is easy to follow.

However I have several concerns:

When bacteria divide, the fluorescence contained in the bacteria divides in the two daughter cells and is therefore halved after cell division and will take some time to return to that of the parent cell. I did not understand how this is taken into account in the model proposed by the authors in the bacterial fluorescence imaging analyses.

Why have the authors not used pyoverdine fluorescence present in the bacteria in order to make a correlation between expression of mcherry and siderophore production. Is this fluorescence of pyoverdine in the cells not strong enough ?

As said by the authors on page 31, the authors only focused on the transcription of the genes involved in siderophore biosynthesis and not on post-transcriptional regulation. The team of P. Arnoux has recently shown a nice regulation of the biosynthesis of staphilopine at the level of the enzyme activity. I am convinced that a regulation exists for all metalloenzyme at the level of the biosynthesis because it is quite expensive to produce large amounts of siderophores and the bacteria need to stop or at least regulate this production at the level of the biosynthesis itself and not just at the level of the transcription of the genes. Nothing has been shown about such a regulation in the case of pyoverdine and pyochelin.

Control by Metals of Staphylopin Dehydrogenase Activity during Metallophore Biosynthesis. Hajjar C, Fanelli R, Laffont C, Brutesco C, Cullia G, Tribout M, Nurizzo D, Borezée-Durant E, Voulhoux R, Pignol D, Lavergne J, Cavelier F, Arnoux P. *J Am Chem Soc.* 2019 Apr 3;141(13):5555-5562. doi: 10.1021/jacs.9b01676.

Consequently, I find it really unfortunate not to have followed the production of siderophores in parallel, especially since in the case of pyoverdine this is extremely easy to follow. Considering the factor impact of Communication Biology and considering the scientific message that the authors propose in this article, it would be essential to have a correlation between the transcription of the genes and the quantity of pyoverdine and pyochelin produced. Pyoverdine production can be followed directly in the culture but for an accurate estimation of pyochelin production the siderophore has to be extracted as in Figure 3A of *Environ Microbiol.* 2015 Jan;17(1):171-85. doi: 10.1111/1462-2920.12544.

Moreover, Line 57, the authors say that : « The regulation of these two siderophores involves three levels. The first level is mediated by Fur (ferric uptake regulator). The second level involves a membrane-embedded signaling cascade, where incoming siderophore-iron complexes trigger a positive feedback loop that increases siderophore production^{27,28}. The third level is pleiotropic in nature.....»

The last years two papers from two different groups have shown that in both, pyoverdine and pyochelin pathways, the positive regulating loop involving the siderophore is more complex as first proposed. The genes encoding the enzymes involved in pyoverdine and pyochelin synthesis can have their transcription activated even in the absence of their corresponding regulators (PvdS and PchR) with as consequence an increase of the production of the siderophores. Considering the topic of the present paper, these two articles need to be cited and discussed.

Interactions between an anti-sigma protein and two sigma factors that regulate the pyoverdine signaling pathway in *Pseudomonas aeruginosa*.

Edgar RJ, Xu X, Shirley M, Konings AF, Martin LW, Ackerley DF, Lamont IL. *BMC Microbiol.* 2014 Nov 30;14:287. doi: 10.1186/s12866-014-0287-2.

The pathogen *Pseudomonas aeruginosa* optimizes the production of the siderophore pyochelin upon environmental challenges.

Cunrath O, Graulier G, Carballido-Lopez A, Pérard J, Forster A, Geoffroy VA, Saint Auguste P, Bumann D, Mislin GLA, Michaud-Soret I, Schalk IJ, Fechter P. *Metallomics*. 2020 Dec 23;12(12):2108-2120. doi: 10.1039/d0mt00029a.

In addition, the difference in the affinities for iron of both siderophores pyoverdine and pyochelin induces a non-equilibrated competition for iron and must affect as well the expression of the proteins of both iron uptake pathways and the production of both siderophores. I am surprised that this fact is not considered in the discussion of the manuscript. In iron restricted condition, in the presence of equivalent concentrations of pyoverdine and pyochelin, iron is mostly chelated by pyoverdine and no pyochelin-iron complex is formed to activate PchR. It is clear that this difference in the affinities for iron of both siderophores impacts the expression levels of the corresponding biosynthetic enzymes and this is not really discussed in the manuscript.

Line 265: "...siderophores, where high pyoverdine inhibits pyochelin synthesis. Although the exact mechanism of this inhibition is unknown, our data suggest that it occurs concomitantly in all cells." These data can be explained by the higher affinity of pyoverdine for iron compared to pyochelin.

Line 248: the authors make the hypothesis that the iron stock in each bacteria can be very different. Are there data in the literature supporting this affirmation or hypothesis ?

Line 364. The authors say that bipyridyl is able to enter bacteria. Is there a reference showing such data. Does it mean that bipyridyl will enter in competition with proteins for metals in bacteria and affect metal homeostasis equilibrium. Is there a risk that bipyridyl picks up the iron which may interact with Fur, that it affects indirectly the Fur regulation ?

Fig 2 and 3 legend, specify that RFU is Relative Fluorescence Unit it may not be obvious to everybody.

Reviewer #3 (Remarks to the Author):

In this work, Mridha and Kummerly investigated siderophore gene expression in *Pseudomonas aeruginosa* at the population and single cell levels. They found that siderophore gene expression is highly heterogeneous in the first phases of growth, while it becomes more homogeneous among cells at later stages of growth. They also observed a positive association in the expression of genes for the two siderophores pyoverdine and pyochelin, and propose that a positive correlation exists between the metabolic state of the cell and siderophore production (mainly in the case of pyoverdine).

Major comments

1. Page 6, lines 111-123. Even assuming that the statistical analysis is correct (it's hard to verify this point from the available data, also considering that the methods used to assess significance are intricate and difficult to follow; see the "Data stitching and statistical analysis" paragraph of the Methods), in my opinion the statement "pyochelin gene expression significantly decreased" (lines 113-144) is inappropriate and misleading, as the decrease in pyochelin gene expression in the presence of iron chelators (especially in the presence of bipyridyl) is not evident at all. According to the data shown in Figures 1 and 2, even if such decrease results statistically significant (using the authors' approach for statistical analysis), it appears to be irrelevant from a biological point of view, especially if compared to the huge increase in pyoverdine gene expression induced by iron chelators. In this view, I also suggest the authors to modify the conclusion of this paragraph, as the results of this population-level analysis do not confirm that "bacteria gradually shift from pyochelin to pyoverdine production when moving from mild to severe iron limitation" (lines 122-123), but suggest that bacteria gradually increase pyoverdine production when moving [...] limitation, while maintaining pyochelin production overall constant.

2. Was the choice of rpsL gene expression as "as a proxy for the overall metabolic activity of a

cell" (lines 206-208) supported by any experimental evidence in this work (in addition to the previous work by Alqarni et al. based on the analysis of gene expression by qRT-PCR under varying carbon source availability)? Considering that the proposed correlation between siderophore production and the metabolic state of the cells is a central conclusion of this manuscript, it would be useful to confirm such choice by testing *rpsL* gene expression at the single cell level and at the population level in cultures treated with increasing concentrations of a growth inhibitory compound (hopefully targeting cell metabolism).

3. The mechanistic basis for the proposed correlation between siderophore gene expression and the metabolic state of the cells (lines 284-290) is not clear and has not been discussed or hypothesized in the Discussion. Considering that the authors are measuring gene expression rather than enzymatic activity, this correlation cannot simply depend on ATP intracellular levels and would imply that there is a regulatory protein or pathway able to induce or repress pyoverdine and/or pyochelin genes in response to the metabolic state of the cell. Can the authors propose any candidates?

4. Page 14, lines 317-327. Honestly, I do not appreciate any coordinated behavior or "decision making" process in the observed trend of siderophore gene expression. The reason why an apparently "coordinated response" emerges at later stages of growth is more likely due to the consumption of the available iron by the bacterial population, which reduces the local (cell to cell) variability in iron availability, rather than by a sort of population- or cell density-dependent regulatory mechanism. I strongly suggest avoiding, or at least mitigating, such speculative conclusions about "information collection" and "decision making" processes.

5. Maybe the authors were not aware of, but a similar analysis has been recently published by Mellini et al. (doi: 10.1128/AEM.02956-20), who used a three-reporter promoter probe plasmid to simultaneously monitor the expression of iron uptake and storage genes (for pyoverdine, pyochelin and bacterioferritin) at single cell level under iron-replete and iron-depleted conditions. It would be appropriate and interesting to discuss whether the data from the present work are in line or not with the Mellini's ones.

Minor points

Lines 183-184. As discussed in the major comment no. 1, I do not agree with this conclusion about the "shift from pyochelin to pyoverdine gene expression". This conclusion should be toned down also in the Discussion (see lines 234-235 and 262-265).

Lines 255-256. This general sentence does not apply to pyochelin signaling, as it does not involve a membrane-bound receptor. Please modify or rephrase the sentence.

Line 276. Although it is the first word of the sentence, *rpsL* cannot be written with the upper case. Please replace "RpsL" with "The *rpsL* gene".

Line 291. Those discussed in the previous paragraph are not "mechanistic explanations", as no mechanistic basis for the proposed correlation between siderophore gene expression and cell metabolism has been described. Please rephrase this sentence.

Lines 298-299. It is not true that "pyoverdine gene expression kicks in later" (with respect to pyochelin gene expression). It simply increases over time, while pyochelin gene expression remains overall constant. In my opinion, there is no evidence in this manuscript supporting the hypothesis that pyochelin is the preferred siderophore during the first stages of growth. This paragraph should be rewritten.

Line 361. Please indicate what dilution was used for the refresh (and/or the initial OD600 of the culture).

Line 365. Please provide a reference for the cell permeability of bipyridyl.

Line 519. Please italicize "*rpsL*".

Legend to Figure 3. Please briefly explain also in this legend how "log-transformed fluorescence values" were calculated and/or what the 0 value represents.

Figures 3, S3 and S4. Change the text in the grey boxes above each graph, as "Iron rich pyochelin/pyoverdine" and similar titles are wrong and confusing.

Figures 2-5. The title of the x and y should be repeated in each panel (for panels A, B, etc).

Figure 5. The inclusion of the lane "Iron" in the y axis, whose title is bipyridyl (μM) or transferrin ($\mu\text{g}/\mu\text{L}$), is incorrect. The authors should think of a more suitable way to show this graph.

Table S1. How was the lag phase determined? If the OD600 is measured with a microtiter plate reader, the lag phase is likely overestimated, as the plate reader cannot detect early growth (very low OD600 values).

We would like to thank the three referees for their constructive comments, which have greatly helped to improve the quality of our paper. In the revised version of our manuscript, we have carefully addressed all concerns raised by the referees, and provide detailed responses to each of the comments. Please find below our detailed responses (in blue) to each of the comments (plain text).

Reviewers' comments:

Reviewer #1 (Remarks to the Author):

1. The aim of this study was to get an insight into the cross-talk and interactions between the two siderophores produced by *Pseudomonas aeruginosa*, pyochelin (PCH, low affinity for iron) and pyoverdine (PVD, high affinity for iron). This work is a follow-up of the nice paper of the same group (Dumas et al. 2013) who showed for the first time that PCH production precedes the PVD production once iron limitation is increasing, but this study was done on batch population, and not at individual cell level, which is the case here. They used to this effect reporter genes PAO1pchEF-mcherry and pvdA mcherry and showed that in batch cultures in CAA iron poor medium the production of PVD increases with the degree of Fe limitation caused by increasing concentrations of dipyrindyl or by transferrin. PCH production quickly plateaued while PVD increases, confirming previous data. At single cell level (an enormous work), using a double reporter (pvdA mcherry-pchEF egfp) they observed first a dual population of PCH producing/non producing cells before the exponential phase, this bimodal expression disappearing during the growth under increasing iron limitation conditions imposed by the presence of transferrin or bipyridyl (figures 3 and 4). They also observed a positive correlation between PCH and PVD gene expression across cells and also between the siderophore genes expression and the expression of the housekeeping rpsL gene, this correlation being weaker for pcheF-rpsL than for pchEF-pvdA. Finally, they present a model explaining their observations where different factors influence the production of the two siderophores, Fur regulation, and the metabolic activity (phase I). In phase II the siderophore-mediated cell signalling regulation is taken into account and in phase III the PVD-mediated PCH repression.

While I agree with the model and find the study well done and quite interesting, still some questions arise, which should be addressed in the discussion.

Response 1: Thank you for a precise summary and positive assessment of our work. Below we have addressed all your comments and explain how we have implemented the necessary changes in the manuscript.

- 1.1 First of all, the problem of PVD-mediated PCH production repression. While it makes sense, there is apparently no explanation for it. One could be that the increased PVD deprives iron from PCH and hence impairs the Fe-PCH binding to the PchR regulator and at the same time causes a decreased availability of the Fe-PCH receptor FptA. Some information can eventually be found in the articles from Michel et al. from the Lausanne group. There is also an old paper from Poole showing that PchR can be a repressor and an activator (like many AraC). But I agree that it is speculative since not much is known about the PchR regulon. It is interesting also to mention that the secondary siderophore quinolobactin of *P. fluorescens* ATCC 17400 production is also repressed by PVD (the QB regulator is also an AraC).

Response 1.1: Thank you for these insightful comments on siderophore regulation. We now explain the regulatory mechanisms in much more detail in the introduction (lines 63-73), and mention the putative pyochelin-inhibition mechanism (lines 75-79).

- 1.2 Line 221: It is true that we know how PCH and PVD production is regulated (Fur, PchR for PCH), Fur-PvdS/FpvI for PVD and its receptor, respectively), but it is more complex than that since in the recent years other regulators have been shown to be involved in the complex regulation of PVD biosynthesis and uptake (OxyR for pvdS)(Qing et al. 2013), other sigma factors are involved as well, such as SigX (Schulz et al., 2015). Very recently there is an interesting article published in ELife where the authors used a Selex approach to determine the targets of all transcription factors (not sigmas) in *P. aeruginosa* and they present in their supplementary figure 9 that 14 regulators are involved in iron uptake regulation (8 for pvdS! while SouR AraC regulator could be involved in the regulation of fpvB (second PVD receptor) and pchR regulation, somehow linking the two siderophores. I just mention this to show that the regulation is quite more complex than expected but fact does not contradict the data obtained in this study since the metabolic activity is likely to be influenced by a plethora of regulators.

Response 1.2: Thank you for highlighting the additional regulatory elements influencing siderophore production. We now refer to this body of work in the introduction (lines 79-81) and discuss their potential effect on single-cell behaviour in the discussion (lines 323-328).

Reviewer #2 (Remarks to the Author):

2. In the Subham Mridha and Rolf Kümmerli paper, the authors followed indirectly the transcription and expression levels of the enzymes involved in the biosynthesis of the siderophores pyoverdine and pyochelin in *P. aeruginosa*. Based on the data obtained concerning these genes transcription and expression, they propose that the chronobiological cycle of siderophore gene expression entails three different phases. The first six hours, both pyochelin and pyoverdine gene expression are highly heterogenous across cells (the majority of cells remains in the off-stage, while a minority of cells shows high siderophore gene expression, mostly pyochelin). In phase 2, heterogeneity in siderophore gene expression is greatly reduced, and all cells switch to an on-state and it coincides with the exponential growth phase. Phase 3 corresponds to a shift in the relative gene expression from pyochelin to pyoverdine, with low heterogeneity across cells. The paper is nicely written and is easy to follow.

Response 2: Thank you for the precise summary and positive assessment of our work. Below we have addressed all your comments and implemented the necessary changes in the manuscript.

However I have several concerns:

2.1 When bacteria divide, the fluorescence contained in the bacteria divides in the two daughter cells and is therefore halved after cell division and will take some time to return to

that of the parent cell. I did not understand how this is taken into account in the model proposed by the authors in the bacterial fluorescence imaging analyses.

Response 2.1: Thank you for highlighting this point. We agree with the reviewer that fluorescence generally decreases after cell division. However, the phenomenon does not bias our results. This is because (i) we did not track individual cells over time, (ii) we imaged very many cells at each time point that (iii) do not divide synchronously. This means that at every single time point we imaged groups of cells that cover the entire spectrum of cell age, from newly divided to old cells just before division. The variation in cell age certainly contributes to the fluorescence heterogeneity observed.

2.2 Why have the authors not used pyoverdine fluorescence present in the bacteria in order to make a correlation between expression of mcherry and siderophore production. Is this fluorescence of pyoverdine in the cells not strong enough?

Response 2.2: This is a very valid point. We have actually collected the pyoverdine fluorescence data in all our single-cell and batch culture experiments. The reason for not including it into the paper was that pyoverdine can be shared between cells and so the fluorescence detected in a cell can stem from both (i) self-produced pyoverdine and (ii) pyoverdine molecules taken up from others. However, this comment motivated us to analyse the data in more detail and to include it into the paper. The batch-culture data yielded strong positive correlations between mCherry fluorescence (capturing pyoverdine gene expression) and actual pyoverdine content in the medium. This provides strong evidence that gene expression is a good proxy for actual siderophore production. At the single-cell level, we recovered the same findings (i.e., strong positive correlations mCherry and pyoverdine fluorescence), but only at the early time points. At later time points, when cells start to share and take up pyoverdine from others, the correlations became weaker, as expected. We now present and discuss these results in the Fig. S5, lines 137-142 and lines 156-162.

2.3 As said by the authors on page 31, the authors only focused on the transcription of the genes involved in siderophore biosynthesis and not on post-transcriptional regulation. The team of P. Arnoux has recently shown a nice regulation of the biosynthesis of staphilopine at the level of the enzyme activity. I am convinced that a regulation exists for all metalloenzyme at the level of the biosynthesis because it is quite expensive to produce large amounts of siderophores and the bacteria need to stop or at least regulate this production at the level of the biosynthesis itself and not just at the level of the transcription of the genes. Nothing has been shown about such a regulation in the case of pyoverdine and pyochelin.

Control by Metals of Staphylopin Dehydrogenase Activity during Metallophore Biosynthesis.

Hajjar C, Fanelli R, Laffont C, Brutesco C, Cullia G, Tribout M, Nurizzo D, Borezée-Durant E, Voulhoux R, Pignol D, Lavergne J, Cavelier F, Arnoux P. *J Am Chem Soc.* 2019 Apr 3;141(13):5555-5562. doi: 10.1021/jacs.9b01676.

Response 2.3: Thank you for this comment. There is in fact tight regulation of siderophore production in response to iron availability. This is supported by our own data showing that none of the two siderophores is expressed under iron-rich conditions. The mechanisms of siderophore synthesis regulation are now explained in much more details in the introduction on lines 63-81 and lines 126+127. Moreover, we have now conducted the correlation analysis between pvdA gene expression and pyoverdine fluorescence (see response 2.2), revealing strong positive relationships.

2.4 Consequently, I find it really unfortunate not to have followed the production of siderophores in parallel, especially since in the case of pyoverdine this is extremely easy to follow. Considering the factor impact of Communication Biology and considering the scientific message that the authors propose in this article, it would be essential to have a correlation between the transcription of the genes and the quantity of pyoverdine and pyochelin produced. Pyoverdine production can be followed directly in the culture but for an accurate estimation of pyochelin production the siderophore has to be extracted as in Figure 3A of Environ Microbiol. 2015 Jan;17(1):171-85. doi: 10.1111/1462-2920.12544.

Response 2.4: Here, we refer to our response 2.2. In the revised version of our paper, we have included data on actual pyoverdine production both at the batch and single-cell level. We further agree that the direct measure of pyochelin is more complicated and even impossible at the single-cell level. We have extracted and measured pyochelin before (Dumas et al. 2013), but large batch volumes are required, which cannot be applied to the single-cell level. But we believe that the very strong data on pyoverdine should be proof enough to demonstrate that gene expression is a good proxy for siderophore production.

2.5 Moreover, Line 57, the authors say that : « The regulation of these two siderophores involves three levels. The first level is mediated by Fur (ferric uptake regulator). The second level involves a membrane-embedded signaling cascade, where incoming siderophore-iron complexes trigger a positive feedback loop that increases siderophore production^{27,28}. The third level is pleiotropic in nature.....». The last years two papers from two different groups have shown that in both, pyoverdine and pyochelin pathways, the positive regulating loop involving the siderophore is more complex as first proposed. The genes encoding the enzymes involved in pyoverdine and pyochelin synthesis can have their transcription activated even in the absence of their corresponding regulators (PvdS and PchR) with as consequence an increase of the production of the siderophores. Considering the topic of the present paper, these two articles need to be cited and discussed.

Interactions between an anti-sigma protein and two sigma factors that regulate the pyoverdine signaling pathway in *Pseudomonas aeruginosa*.

Edgar RJ, Xu X, Shirley M, Konings AF, Martin LW, Ackerley DF, Lamont IL. BMC Microbiol. 2014 Nov 30;14:287. doi: 10.1186/s12866-014-0287-2.

The pathogen *Pseudomonas aeruginosa* optimizes the production of the siderophore pyochelin upon environmental challenges.

Cunrath O, Graulier G, Carballido-Lopez A, Pérard J, Forster A, Geoffroy VA, Saint Auguste P, Bumann D, Mislin GLA, Michaud-Soret I, Schalk IJ, Fechter P. Metallomics. 2020 Dec 23;12(12):2108-2120. doi: 10.1039/d0mt00029a.

Response 2.5: We appreciate this comment. A similar comment has been raised by reviewer #1 (see our response 1.2). We agree that these novel insights are important. We now mention these additional papers and regulatory elements in the introduction (lines 79-81) and discuss their implications in the discussion (lines 323-328).

2.6 In addition, the difference in the affinities for iron of both siderophores pyoverdine and pyochelin induces a non-equilibrated competition for iron and must affect as well the

expression of the proteins of both iron uptake pathways and the production of both siderophores. I am surprised that this fact is not considered in the discussion of the manuscript. In iron restricted condition, in the presence of equivalent concentrations of pyoverdine and pyochelin, iron is mostly chelated by pyoverdine and no pyochelin-iron complex is formed to activate PchR. It is clear that this difference in the affinities for iron of both siderophores impacts the expression levels of the corresponding biosynthetic enzymes and this is not really discussed in the manuscript.

Response 2.6: This is a very good point. Reviewer #1 offered a similar explanation (see response 1.1). We now discuss this regulatory link on lines 75-79 and 296-302.

2.7 Line 265: "...siderophores, where high pyoverdine inhibits pyochelin synthesis. Although the exact mechanism of this inhibition is unknown, our data suggest that it occurs concomitantly in all cells." These data can be explained by the higher affinity of pyoverdine for iron compared to pyochelin.

Response 2.7: This relates to comment 2.6 and we have now addressed this point in detail in the manuscript. Please check our response 2.6.

2.8 Line 248: the authors make the hypothesis that the iron stock in each bacteria can be very different. Are there data in the literature supporting this affirmation or hypothesis?

Response 2.8: Thank you for this comment. We are not aware of a study that examined single-cell variation in internal iron stores. However, we have now added more detailed information on the mechanism of internal iron storage and explain how inter-cell variation could arise (lines 284-291).

2.9 Line 364. The authors say that bipyridyl is able to enter bacteria. Is there a reference showing such data. Does it mean that bipyridyl will enter in competition with proteins for metals in bacteria and affect metal homeostasis equilibrium. Is there a risk that bipyridyl picks up the iron which may interact with Fur, that it affects indirectly the Fur regulation?

Response 2.9: This is a good question. The exact mechanism by which bipyridyl affects bacteria is not known (to our knowledge). The assumption is that bipyridyl indeed interferes with iron homeostasis. But some general (iron-independent) toxicity has also been demonstrated. We now refer to the relevant literature on lines 415+416.

2.10 Fig 2 and 3 legend, specify that RFU is Relative Fluorescence Unit it may not be obvious to everybody.

Response 2.10: We implemented the changes in the figure legends 1-3 as requested.

Reviewer #3 (Remarks to the Author):

3. In this work, Mridha and Kummerly investigated siderophore gene expression in

Pseudomonas aeruginosa at the population and single cell levels. They found that siderophore gene expression is highly heterogeneous in the first phases of growth, while it becomes more homogeneous among cells at later stages of growth. They also observed a positive association in the expression of genes for the two siderophores pyoverdine and pyochelin, and propose that a positive correlation exists between the metabolic state of the cell and siderophore production (mainly in the case of pyoverdine).

Response 3. Thank you for a precise summary of our work. Below we addressed all your comments and implemented the necessary changes in the manuscript.

Major comments

3.1 Page 6, lines 111-123. Even assuming that the statistical analysis is correct (it's hard to verify this point from the available data, also considering that the methods used to assess significance are intricate and difficult to follow; see the "Data stitching and statistical analysis" paragraph of the Methods), in my opinion the statement "pyochelin gene expression significantly decreased" (lines 113-144) is inappropriate and misleading, as the decrease in pyochelin gene expression in the presence of iron chelators (especially in the presence of bipyridyl) is not evident at all. According to the data shown in Figures 1 and 2, even if such decrease results statistically significant (using the authors' approach for statistical analysis), it appears to be irrelevant from a biological point of view, especially if compared to the huge increase in pyoverdine gene expression induced by iron chelators. In this view, I also suggest the authors to modify the conclusion of this paragraph, as the results of this population-level analysis do not confirm that "bacteria gradually shift from pyochelin to pyoverdine production when moving from mild to severe iron limitation" (lines 122-123), but suggest that bacteria gradually increase pyoverdine production when moving [...] limitation, while maintaining pyochelin production overall constant.

Response 3.1. Thank you for this comment. We understand the concerns. That is why we conducted additional analysis to examine the change in pyochelin vs. pyoverdine gene expression across the gradient of iron limitation and time.

- We re-analysed the batch culture data shown in Figure 1. Specifically, we plotted individual data points for each bipyridyl and transferrin concentration and used regression analysis (the appropriate statistical approach) to test whether pyochelin and pyoverdine gene expression significantly change over time. These analyses confirm our previous observations that pyochelin significantly drops, while pyoverdine significantly increases with stronger levels of iron limitation. We now present these new analyses in the new Fig. S4.
- The reviewer argues that the decline in pyochelin expression is biological irrelevant. We disagree with this view. This impression might have arisen because the promoter strength for pyochelin was weaker compared to pyoverdine. But if we quantify the observed change in percentage (in the new Fig. S4), we find that pyochelin gene expression decreased by up to ~35%, while pyoverdine increased by up to ~40% across the transferrin gradient. This is substantial and that is why we consider it as biologically relevant.
- We also reanalysed the temporal single-cell gene expression data in Fig. 3. We agree that the initial analysis in the former Fig. S4 was confusing. We removed this figure from the paper. Instead, we now examine the change in gene expression from time point 9 hours onwards when all cells have switched on gene expression. We found that under all conditions of iron limitation (six out of six cases), pyochelin gene expression

declined over time, despite the high heterogeneity observed among cells. This confirms our batch culture data that pyochelin production indeed declines over time.

- Based on the new analysis, we revised the text in the manuscript. We agree with the reviewer that there was never a complete shift from pyochelin to pyoverdine production, but rather a change in the relative investment. We emphasize this very clearly in the results and discussion section. Furthermore, we revised the text throughout the manuscript to highlight that the transition was moderate, yet it occurred. Finally, we now also offer a mechanistic explanation for the transition, based on the helpful comments of reviewers #1 and #2 (see our responses 1.1 and 2.6).

3.2 Was the choice of *rpsL* gene expression as “as a proxy for the overall metabolic activity of a cell” (lines 206-208) supported by any experimental evidence in this work (in addition to the previous work by Alqarni et al. based on the analysis of gene expression by qRT-PCR under varying carbon source availability)? Considering that the proposed correlation between siderophore production and the metabolic state of the cells is a central conclusion of this manuscript, it would be useful to confirm such choice by testing *rpsL* gene expression at the single cell level and at the population level in cultures treated with increasing concentrations of a growth inhibitory compound (hopefully targeting cell metabolism).

Response 3.2. We appreciate this comment. We have conducted the suggested experiment by using trimethoprim as a metabolic inhibitor. The results of this new experiment are presented on lines 233-247 and in Fig. S9, discussed on lines 331-337, and the methods presented on lines 442-458. The results were in support of our assumption that *rpsL* gene expression is a good proxy for metabolic activity.

3.3 The mechanistic basis for the proposed correlation between siderophore gene expression and the metabolic state of the cells (lines 284-290) is not clear and has not been discussed or hypothesized in the Discussion. Considering that the authors are measuring gene expression rather than enzymatic activity, this correlation cannot simply depend on ATP intracellular levels and would imply that there is a regulatory protein or pathway able to induce or repress pyoverdine and/or pyochelin genes in response to the metabolic state of the cell. Can the authors propose any candidates?

Response 3.3. We apologize for our imprecise phrasing. We were not thinking of a particular protein, but phrased our argument in a conceptual way. It is very simple. If a cell has low metabolic activity, we assume that the expression of all important genes (e.g. housekeeping genes and genes involved in iron acquisition) is reduced. Conversely, if a cell has high metabolic activity, we assume that the expression of all important genes is increased. Across cells varying in their metabolic activity, we would expect positive correlations between any two genes (e.g. house-keeping and siderophore genes). This concept holds regardless of the underlying mechanisms. This is what we aimed to test here. We have now improved the explanations on our approach on lines 238-241.

3.4 Page 14, lines 317-327. Honestly, I do not appreciate any coordinated behavior or “decision making” process in the observed trend of siderophore gene expression. The reason why an apparently “coordinated response” emerges at later stages of growth is more likely due to the consumption of the available iron by the bacterial population, which reduces the local (cell to cell) variability in iron availability, rather than by a sort of population- or cell density-dependent regulatory mechanism. I strongly suggest avoiding, or at least mitigating, such speculative conclusions about “information collection” and “decision making” processes.

Response 3.4. We understand this criticism. Our aim is not to use jargon to crave attention. However, we think it is okay to critically discuss biological concepts developed for higher organisms in the context of microbes. Collective decision-making and information collection does not necessarily imply cognitive abilities. Seminal work by Couzin and colleagues showed that group coordination can arise through very simple mechanisms: individuals collect information from their local neighbours and align their behaviour. Now it is well established that quorum-sensing in bacteria follows this principle: individuals receive signals from their neighbours and coordinate responses. We now apply the exact same principles to siderophores: bacteria take up ferri-siderophores from the neighbourhood, which triggers intra-cellular signalling and siderophore synthesis. Since siderophores are shared between, signalling is supposed to become more homogenous with higher cell densities, which should homogenize behaviour across cells. This is exactly what we observe. In our view, this is not a speculation but the application of a well-established concept to our study system. To clarify this point, we have revised our arguments on lines 370-379.

3.5 Maybe the authors were not aware of, but a similar analysis has been recently published by Mellini et al. (doi: 10.1128/AEM.02956-20), who used a three-reporter promoter probe plasmid to simultaneously monitor the expression of iron uptake and storage genes (for pyoverdine, pyochelin and bacterioferritin) at single cell level under iron-replete and iron-depleted conditions. It would be appropriate and interesting to discuss whether the data from the present work are in line or not with the Mellini's ones.

Response 3.5. Thank you for bringing this paper to our attention. We now cite it in our revised paper to highlight it as an additional tool (lines 402-404). However, it is difficult to directly compare the two papers, as Mellini et al. primarily report qualitative measures on the percentage of fluorescent cells. There are no quantitative single-cell analysis as in our paper.

Minor points

3.6 Lines 183-184. As discussed in the major comment no. 1, I do not agree with this conclusion about the “shift from pyochelin to pyoverdine gene expression”. This conclusion should be toned down also in the Discussion (see lines 234-235 and 262-265).

Response 3.6. As explained in our response 3.1 above, we have reanalysed our data to examine the changes in gene expression over time and across environments in more detail. Accordingly, we have also revised our conclusions in the main text.

3.7 Lines 255-256. This general sentence does not apply to pyochelin signaling, as it does not involve a membrane-bound receptor. Please modify or rephrase the sentence.

Response 3.7. Thank you for specifying this aspect. We have corrected this mistake on lines 69-73 and lines 302-305.

3.8 Line 276. Although it is the first word of the sentence, rpsL cannot be written with the upper case. Please replace “RpsL” with “The rpsL gene”.

Response 3.8. We have revised the sentence as suggested.

3.9 Line 291. Those discussed in the previous paragraph are not “mechanistic explanations”, as no mechanistic basis for the proposed correlation between siderophore gene expression and cell metabolism has been described. Please rephrase this sentence.

Response 3.9. Thank you for this comment. We have revised the start of this section and have removed the term “mechanistic explanations” (lines 345-347).

3.10 Lines 298-299. It is not true that “pyoverdine gene expression kicks in later” (with respect to pyochelin gene expression). It simply increases over time, while pyochelin gene expression remains overall constant. In my opinion, there is no evidence in this manuscript supporting the hypothesis that pyochelin is the preferred siderophore during the first stages of growth. This paragraph should be rewritten.

Response 3.10. We agree that our description of the data was imprecise. This comment goes along with point 3.1, where we explain in our response that we have changed our argumentation on the temporal pyoverdine versus pyochelin investment. In this context, we have also removed the statement criticized above.

3.11 Line 361. Please indicate what dilution was used for the refresh (and/or the initial OD600 of the culture).

Response 3.11. All our experiments were started with an initial OD600 = 0.0001. We have clarified this throughout the method section.

3.12 Line 365. Please provide a reference for the cell permeability of bipyridyl.

Response 3.12. As per request, we provide a reference for bipyridyl cell permeability (line 416).

3.13 Line 519. Please italicize “rpsL”.

Response 3.13. Done as per request.

3.14 Legend to Figure 3. Please briefly explain also in this legend how “log-transformed fluorescence values” were calculated and/or what the 0 value represents.

Response 3.14. We have added the requested explanations to the legend of Figure 3.

3.15 Figures 3, S3 and S4. Change the text in the grey boxes above each graph, as “Iron rich pyochelin/pyoverdine” and similar titles are wrong and confusing.

Response 3.15. Thank you for highlighting the misleading labels. We have checked and corrected the titles in all the figures mentioned above.

3.16 Figures 2-5. The title of the x and y should be repeated in each panel (for panels A, B, etc).

Response 3.16. We understand the reviewer's request. However, it is not uncommon to indicate the axes titles only once in cases where all panels have the same titles. Most of our figures busy and that is why we prefer to reduce the number of repetitions for axes titles.

3.17 Figure 5. The inclusion of the lane "Iron" in the y axis, whose title is bipyridyl (μM) or transferrin ($\mu\text{g}/\mu\text{L}$), is incorrect. The authors should think of a more suitable way to show this graph.

Response 3.17. We agree that these labels are confusing. We have corrected them, so that each row in the heatmap comes with a clear label.

3.18 Table S1. How was the lag phase determined? If the OD600 is measured with a microtiter plate reader, the lag phase is likely overestimated, as the plate reader cannot detect early growth (very low OD600 values).

Response 3.18. We agree that this a valid point. All growth data stem from plate reader experiments. We fitted parametric mathematical models to estimate the growth parameters. We agree that the plate reader detection limit prevents to measure the first cell divisions. So, the lag-phase we are estimating here reflects the time point at which the culture surpasses the detection limit of the plate reader. Since all cultures undergo the exact same treatment, it is valid to use the plate reader lag-time and to compare it across treatments/conditions, even when it might not exactly reflect the lag-time as one would measure it with another method (i.e. CFU on plates). We have now clarified this in a footnote of Table S1.

REVIEWERS' COMMENTS:

Reviewer #1 (Remarks to the Author):

The manuscript from Mridha et al. has been revised in accordance with the three reviewers comments, which have for the majority of the cases correctly addressed. I am therefore satisfied with the changes made in the manuscript, including the supplementary material.

Reviewer #2 (Remarks to the Author):

The authors have broadly answered my questions and comments. However, before publication, an error needs to be corrected. Concerning the affinity for iron of PCH, the authors give the following value: $K_a = 2 \times 10^5 \text{ M}^{-1}$ (line 77). This affinity has been determined in (ethanol). The authors must use the affinity determined in aqueous solutions $K_a = 1018 \text{ M}^{-2}$, published in doi: 10.1039/c1dt11804h. M^{-2} because ferric iron is chelated by two molecules of pyochelin.

Reviewer #3 (Remarks to the Author):

I want to congratulate the authors for clarifying the critical points as best as possible and enriching the manuscript with the required controls.